# Graph pangenome captures missing heritability and empowers tomato breeding

Yao Zhou[1,13], Zhiyang Zhang[1,13], Zhigui Bao[1,13], Hongbo Li[1], Yaqing Lyu[1], Yanjun Zan[1,2], Yaoyao Wu[1], Lin Cheng[1], Yuhan Fang[1], Kun Wu[1], Jinzhe Zhang[3], Hongjun Lyu[1,4], Tao Lin[5], Qiang Gao[6], Surya Saha[7], Lukas Mueller[7], Zhangjun Fei[7,8], Thomas Städler[9], Shizhong Xu[10], Zhiwu Zhang[11], Doug Speed[12] & Sanwen Huang[1✉]

Missing heritability in genome-wide association studies defines a major problem in genetic analyses of complex biological traits[1,2]. The solution to this problem is to identify all causal genetic variants and to measure their individual contributions[3,4]. Here we report a graph pangenome of tomato constructed by precisely cataloguing more than 19 million variants from 838 genomes, including 32 new reference-level genome assemblies. This graph pangenome was used for genome-wide association study analyses and heritability estimation of 20,323 gene-expression and metabolite traits. The average estimated trait heritability is 0.41 compared with 0.33 when using the single linear reference genome. This 24% increase in estimated heritability is largely due to resolving incomplete linkage disequilibrium through the inclusion of additional causal structural variants identified using the graph pangenome. Moreover, by resolving allelic and locus heterogeneity, structural variants improve the power to identify genetic factors underlying agronomically important traits leading to, for example, the identification of two new genes potentially contributing to soluble solid content. The newly identified structural variants will facilitate genetic improvement of tomato through both marker-assisted selection and genomic selection. Our study advances the understanding of the heritability of complex traits and demonstrates the power of the graph pangenome in crop breeding.

Missing heritability—the discrepancy between heritability estimates from family-based genetic studies and the variance explained by all of the significant variants in genome-wide association studies (GWAS)[1,2]—compromises the use of rapidly developing genomics for understanding biological questions and crop breeding[5–7]. The resolution of missing heritability is hindered by several factors, including incomplete detection of causal genomic variants, particularly structural variants (SVs), which leads to estimation bias caused by incomplete linkage disequilibrium (LD) between genetic markers and causal variants, as well as genetic heterogeneity of causal variants, which reduces the statistical power of GWAS[8–10]. To overcome these bottlenecks, an exhaustive and precise catalogue of genetic variants is required.

A variation map constructed by mapping sequencing reads to a single linear reference genome generates reference bias, that is, the inability to precisely map non-reference alleles[11,12]. A pangenome comprising multiple reference genomes may more fully represent species-wide genetic diversity and, as such, retains non-reference information[13–15].

However, it is challenging to incorporate coordinates of non-reference sequences into existing analysis pipelines[16]. Recently, graph-based structures have been used to integrate all genetic variants into a single genome graph, enabling thorough and accurate identification of genomic variants as well as data integration[11,17–19]. Recent studies have demonstrated the superiority of using graph pangenomes as references in identification of SVs with short reads[19–22]. Here we report the construction of a variant-based graph pangenome of tomato (*Solanum lycopersicum*), an important fruit crop and a model system for plant biology and breeding. We demonstrate its use in capturing missing heritability in GWAS, providing insights into a classical genetics problem and facilitating genomic breeding (Extended Data Fig. 1).

## Construction of the graph pangenome

A high-accuracy and gapless linear reference genome is as critical as the backbone of a graph pangenome. To this end, we assembled a

[1]Shenzhen Branch, Guangdong Laboratory of Lingnan Modern Agriculture, Genome Analysis Laboratory of the Ministry of Agriculture and Rural Affairs, Agricultural Genomics Institute at Shenzhen, Chinese Academy of Agricultural Sciences, Shenzhen, China. [2]Umeå Plant Science Center, Department of Forestry Genetics and Plant Physiology, Swedish University of Agricultural Sciences, Umeå, Sweden. [3]Key Laboratory of Biology and Genetic Improvement of Horticultural Crops of the Ministry of Agriculture, Sino-Dutch Joint Laboratory of Horticultural Genomics, and Institute of Vegetables and Flowers, Chinese Academy of Agricultural Sciences, Beijing, China. [4]Institute of Vegetables, Shandong Academy of Agricultural Sciences, Shandong Province Key Laboratory for Biology of Greenhouse Vegetables, Shandong Branch of National Improvement Center for Vegetables, Huang-Huai-Hai Region Scientific Observation and Experimental Station of Vegetables, Ministry of Agriculture and Rural Affairs, Jinan, China. [5]State Key Laboratory of Agrobiotechnology, College of Horticulture, China Agricultural University, Beijing, China. [6]Boke Biotech, Wuxi, China. [7]Boyce Thompson Institute, Cornell University, Ithaca, NY, USA. [8]Robert W. Holley Center for Agriculture and Health, US Department of Agriculture, Agricultural Research Service, Ithaca, NY, USA. [9]Institute of Integrative Biology & Zurich, Basel Plant Science Center, ETH Zurich, Zurich, Switzerland. [10]Department of Botany and Plant Sciences, University of California, Riverside, CA, USA. [11]Department of Crop and Soil Sciences, Washington State University, Pullman, WA, USA. [12]Quantitative Genetics and Genomics (QGG), Aarhus University, Aarhus, Denmark. [13]These authors contributed equally: Yao Zhou, Zhiyang Zhang, Zhigui Bao. ✉e-mail: huangsanwen@caas.cn

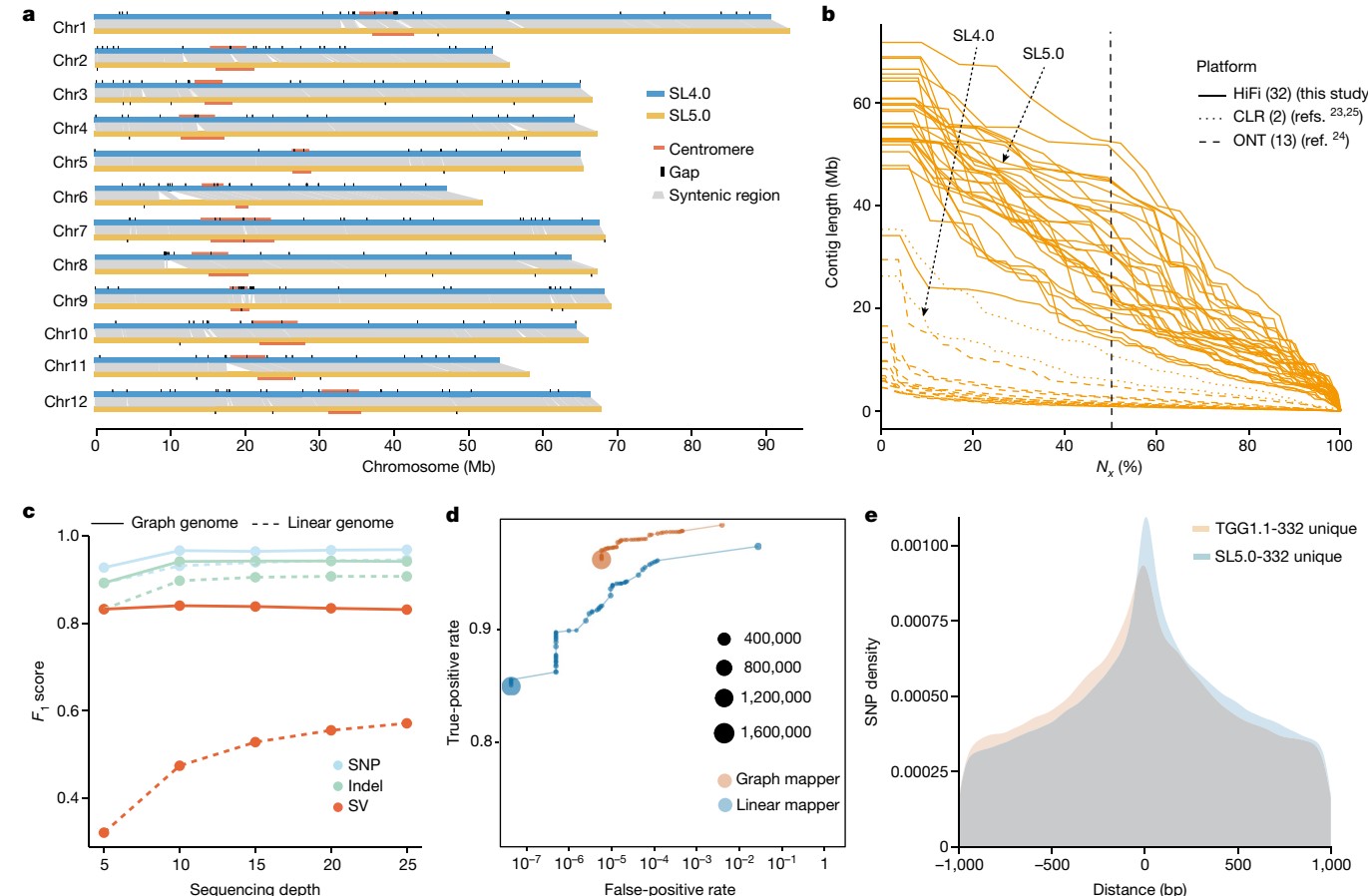

**Fig. 1 | Genome and graph pangenome of tomato. a**, Synteny between tomato reference genome build SL4.0 (blue) and SL5.0 (yellow). The grey lines represent synteny blocks. The positions of gaps are marked with black rectangles on the chromosomes, and centromeres are represented by orange rectangles along the chromosomes. **b**, Contig $N_x$ size of all genome assemblies. SL4.0 and SL5.0 are marked with arrows. Line types represent different sequencing platforms. CLR, PacBio continuous long reads; HiFi, high-fidelity long reads; ONT, Oxford Nanopore long reads. The numbers in parentheses refer to the numbers of assemblies. **c**, $F_1$ scores (harmonic means

of precision and recall) using simulated sequencing data from the genetic variants of 31 accessions with HiFi reads with different depths and genetic variants from the graph pangenome and the linear genome. **d**, Assessing false-positive (*x*-axis) and true-positive (*y*-axis) rates for the graph (Giraffe) and linear (BWA-MEM) mappers using 2,000,000 simulated reads. The size of each point represents the number of reads with mapping quality equal to 60. **e**, Density map of unique SNPs from SL5.0-332 and TGG1.1-332 located within 1 kb upstream or downstream of the SV breakpoints.

state-of-the-art backbone genome (tomato cv. Heinz 1706, Build SL5.0) using high-fidelity (HiFi) long reads and high-throughput chromosome conformation capture (Hi-C) long-range scaffolding (Extended Data Fig. 2a). The contig $N_{50}$ size of SL5.0 is 41.7 Mb, an increase of approximately sevenfold compared with the previous build SL4.0 (ref. [23]). Moreover, SL5.0 contains 19.3 Mb more sequences than SL4.0 (801.8 Mb versus 782.5 Mb), with 43 contigs (99.8% of the assembly) ordered and oriented on the 12 chromosomes (Fig. 1a and Extended Data Fig. 2b). Only 31 gaps remain in the SL5.0 pseudochromosomes, substantially fewer than in SL4.0 (259 gaps). Gaps remain mostly in highly complex regions, including subtelomeres, centromeres and rDNA repeats. Both bacterial artificial chromosome clone sequences and *k*-mer analysis support the superior quality of SL5.0 (Supplementary Table 1). We performed the annotation of SL5.0 (ITAG5.0), predicting 36,648 protein-coding genes.

We generated reference-level genome assemblies for another 31 accessions that represent the diversity of the red-fruited clade of tomatoes, including 15 big-fruited tomato *S. lycopersicum* (BIG) accessions, eight cherry tomato (*S. lycopersicum* var. *cerasiforme*, CER) accessions and eight accessions from *S. pimpinellifolium* (PIM, considered to be the progenitor of cultivated tomatoes) (Supplementary Table 2 and Supplementary Fig. 1). The contig $N_{50}$ sizes of these 31 assemblies range from 13.7 Mb to 52.2 Mb, with an average of 28.6 Mb, larger than any

of the previously published tomato pangenome assemblies MAS2.0 (ref. [24]) (Fig. 1b and Supplementary Table 3). We annotated repeats and predicted protein-coding genes for 45 assemblies: 31 from this study, 13 from MAS2.0 (eight BIG, three CER and two PIM accessions)[24] and 1 PIM accession from another study[25]. The content of repetitive sequences ranges from 60.7% to 64.0%, with an average of 62.1% (Supplementary Table 4). The number of predicted protein-coding genes ranges from 33,863 to 37,237, with an average of 35,298 (Supplementary Table 5). The completeness of these assemblies was assessed by BUSCO analysis, which shows an average of 96.2% single-copy Solanales genes completely assembled (Extended Data Fig. 2c). Taken together, these high-quality genome assemblies represent a robust resource to facilitate variant detection and genomic comparison for constructing a tomato graph pangenome.

With SL5.0 serving as the backbone, single-nucleotide polymorphisms (SNPs) and small insertions and deletions (indels, 1–50 bp) identified from the 31 accessions with HiFi reads, as well as SVs (>50 bp) from all 131 accessions with long reads (a total of 100 accessions from a previous study[24] and 31 accessions from this study), were integrated into a variation graph. Complex SVs were not specifically considered when constructing the graph pangenome (Supplementary Note 4). The resulting tomato graph pangenome (TGG1.0) spans 1,007,562,373 bp, including approximately 206 Mb

absent from SL5.0. We mapped all predicted protein-coding genes to a graph generated from all assemblies, resulting in a tomato graph annotation (TGA1.0) with 51,155 genes, of which 14,507 are from the non-reference genomes. Previous resequencing projects accumulated 7.8 Tb of Illumina short-read data for 706 tomato accessions with a sequencing depth of greater than sixfold[26–31]. By mapping these short reads to TGG1.0, we identified additional SNPs and indels that were not present in TGG1.0. After merging these variants with those from TGG1.0, we obtained a dataset comprising 17,898,731 SNPs, 1,499,161 indels and 195,957 SVs. Integration of this updated genetic variant dataset and the SL5.0 backbone genome resulted in the generation of a new variation graph, which we designate TGG1.1.

Simulation studies indicate that the graph pangenome outperforms the linear genome at calling all types of genetic variants (SNPs, indels and SVs) (Supplementary Table 6), consistent with a recent study on a human variation graph[12,19]. We compared the performance metrics for SNPs, indels and SVs derived from the graph pangenome and the linear genome. From the raw output of genotypes, we obtained $F_1$ scores (harmonic mean of precision and recall) of 0.966 for SNPs, 0.941 for indels and 0.840 for SVs in the graph pangenome using 10× sequencing data, significantly better than those in the linear genome (0.931, 0.897 and 0.474; Wilcoxon rank sum test, $P = 6.30 \times 10^{-13}$, $P = 5.04 \times 10^{-14}$ and $P = 1.69 \times 10^{-17}$, respectively) (Fig. 1c). Given that the same variant caller DeepVariant[32] was used for both datasets, higher precision and recall rate is probably driven by the higher accuracy of mapping short reads using the graph mapper (Fig. 1d).

Next, we genotyped genetic variants of 332 tomato accessions by mapping their Illumina sequences onto TGG1.1, resulting in a callset designated TGG1.1-332 that comprises 6,971,059 SNPs, 657,549 indels and 54,838 SVs. We also mapped these sequences against the linear genome SL5.0 and identified variants in a callset designated SL5.0-332 comprising 7,317,844 SNPs, 447,098 indels and 11,397 SVs. We found that SNPs that were uniquely identified by the linear reference were physically closer to their neighbouring SVs than SNPs uniquely identified by the graph pangenome (Fig. 1e), consistent with lower levels of incorrect read mapping around SVs in the latter dataset (Extended Data Fig. 3). Furthermore, TGG1.1 contains 7,197 out of the 7,720 SNPs (93.2%) that were verified in a DNA chip[33], whereas only 6,812 (88.2%) were detected using SL5.0 as the reference. Notably, the linear genome yields only 20% of the SVs called by the graph pangenome, indicating the high efficiency in detecting SVs using the graph pangenome. In summary, TGG1.1 represents one of the most comprehensive and accurate maps of tomato genome variation to date.

## Capturing missing heritability

To test the power of the graph pangenome in capturing missing heritability, we used LDAK[34] to estimate the variant heritability of 20,323 molecular traits, comprising 19,353 expression traits and 970 metabolite traits, from fruits of the 332 tomato accessions[35]. First, we analysed each category of genetic variants individually (that is, only SNPs, only indels or only SVs). The average heritability estimated using the graph pangenome is higher than that using the linear reference genome for all three categories (Fig. 2a and Supplementary Table 7). Higher SNP heritability (0.29 versus 0.28; Wilcoxon rank sum test, $P = 7.24 \times 10^{-3}$; Extended Data Fig. 4b) is suggested despite TGG1.1-332 comprising fewer SNPs than SL5.0-332. The results were similar when this analysis was restricted to 6,375 independent traits (square of Pearson's correlation coefficient ($r^2$) between the traits, <0.20) (Extended Data Fig. 4a).

We next analysed categories of genetic variants jointly. Estimated heritability increases with more categories in the model (Fig. 2a). When jointly analysing all three categories of variants in a composite model, the average heritability is 0.41 in the graph pangenome callset, 24% higher than that in the linear genome callset (0.33; Wilcoxon rank

sum test, $P = 1.23 \times 10^{-217}$). We used the composite model to estimate the average heritability explained by SNPs, indels and SVs from TGG1.1-332, finding that SVs contribute the largest proportion of overall heritability (0.27, 65.9%) (Extended Data Fig. 4c). Moreover, SVs contribute the largest share of heritability for approximately half of the molecular traits (10,297 out of 20,323, 50.7%) (Fig. 2b). These data indicate that the capture of missing heritability through the graph pangenome is largely due to the inclusion of more identified SVs.

Incomplete LD between molecular markers and causal variants leads to the underestimation of heritability[9]. SVs in close proximity to genes are probably causal variants as they could lead to dysregulation of gene expression[24,36]. We observed that a large proportion of SVs are in strong LD ($R^2 > 0.7$) with adjacent (50 kb on either side) SNPs and indels (61.2% and 45.5%, respectively), but only small fractions (3.2% and 0.6%, respectively) are in complete LD ($R^2 = 1$) (Fig. 2c), indicating that incomplete LD between markers and causal variants is common in our population. Our simulation studies show that inclusion of causal variants captures some missing heritability (Supplementary Fig. 2). This could, at least partially, explain why the average heritability increases from 0.37 to 0.41 when SVs are included in the model compared with models that consider only SNPs and indels (Fig. 2a).

As an example, we studied the case of *Solyc03G002957*, which encodes a protein that interacts with phosphoinositides. To evaluate the effects of *cis*-variants on gene expression, we partitioned genetic variants into six categories, namely *cis*-variants (50 kb on either side of the gene) and *trans*-variants of SNPs, indels and SVs from the linear and graph pangenome callset, respectively. We found that total heritability estimated from SL5.0-332 is 0.54 (s.d. = 0.32). By contrast, total heritability estimated from TGG1.1-332 is 0.75 (s.d. = 0.51), to which *cis*- and *trans*-SVs jointly contribute the largest proportions, 0.41 (s.d. = 0.34) and 0.28 (s.d. = 0.10), respectively (Fig. 2d). This indicates that SVs around this gene, most of which can be identified only using the graph-based approach, are more likely to be causative than other variant types and contribute to the majority of total heritability.

When we performed a single-variant association study, we found that the expression of *Solyc03G002957* is probably affected by a SV, a leading variant residing at a peak on chromosome 3 (sv3_62128422, a 2,628 bp deletion causing a truncation at the end of the transcript) (Fig. 2e and Extended Data Figs. 5 and 6). This SV explains approximately 0.45 (s.d. = 0.63) of heritability and is present only in TGG1.1-322. However, a significant SNP (SNP3_62204487, located about 57.6 kb upstream from the gene) exhibits modest LD with the SV ($R^2 = 0.66$) (Fig. 2e) and explains 0.34 (s.d. = 0.48) of heritability in both SL5.0-332 and TGG1.1-332. However, given the fact that SNP3_62204487 is eight genes away from the target gene, the statistical significance of this SNP could give misleading results. These results suggest that, by addressing incomplete LD through inclusion of possibly causal SVs, the graph pangenome has the potential to capture missing heritability.

A marked discrepancy still exists between the estimated heritability and the heritability explained by GWAS significant loci[2]. One of the important sources is allelic heterogeneity (that is, multiple underlying genetic variants at the same locus contribute to the same phenotype), a widespread phenomenon in complex traits that tends to impair the power of GWAS[37,38]. To assess the potential effect of allelic heterogeneity on GWAS in tomato, we analysed the effects of variants in *cis*-regions (within 50 kb on either side of genes) on their corresponding gene expression (19,353 genes). Using a single-locus mixed linear model (MLM)[39] on the TGG1.1-332 callset, we detected *cis*-expression quantitative trait loci (eQTLs) for 1,179 genes. Although the average estimated heritability of the expression of these genes is 0.62, the average heritability explained by leading significant variants is only 0.27 (Fig. 3a). Thus, heritability contributed by nearby genetic variants might be 'invisible' when considering only leading significant variants within eQTLs. When including all genetic variants in *cis*-regions of eQTLs (within 50 kb on either side of the leading variant), the average estimated

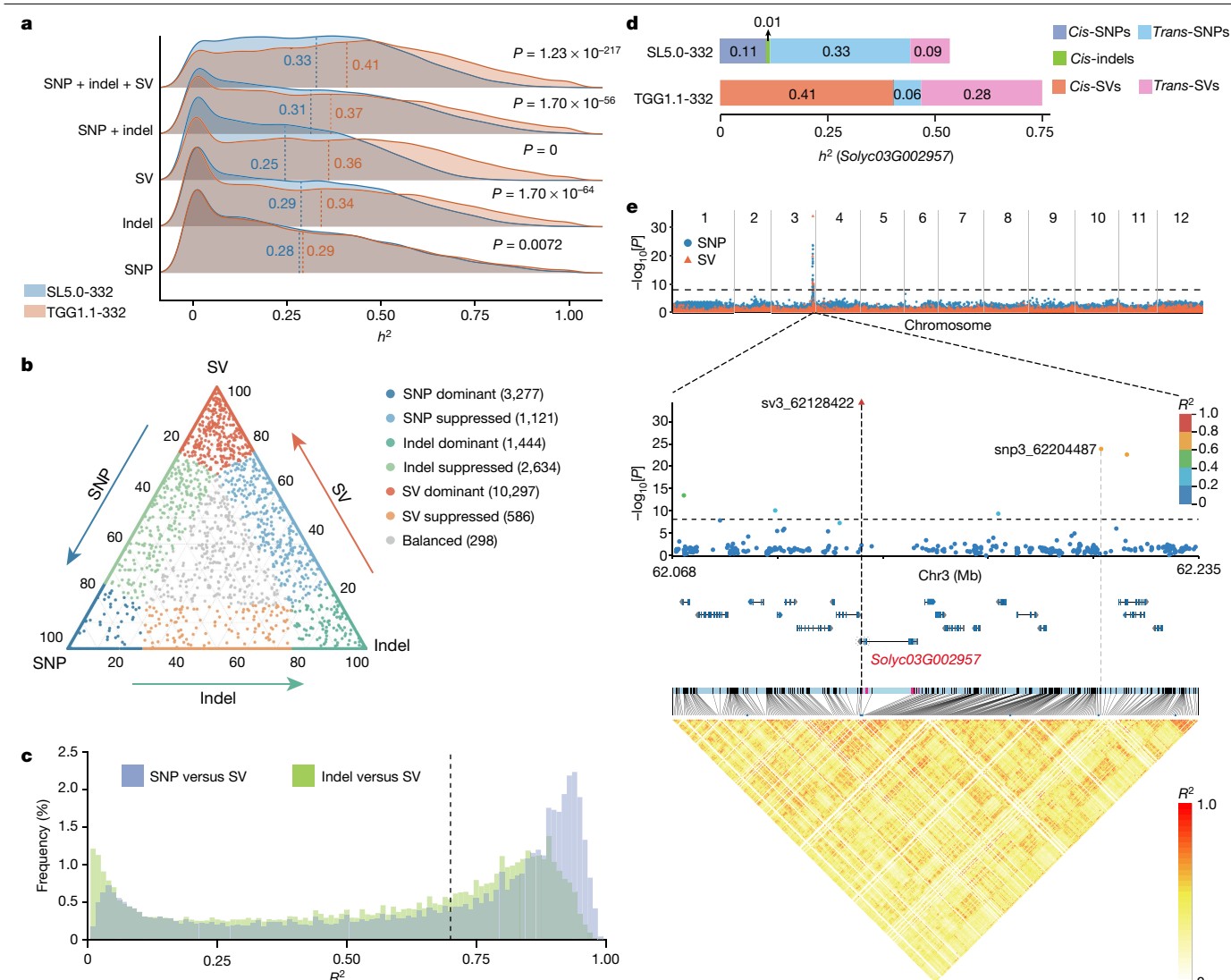

**Fig. 2 | The contribution of genetic variants to heritability. a**, Comparison of heritability ($h^2$) estimated using different combinations of genetic variants from SL5.0-332 and TGG1.1-332. SNP + indel and SNP + indel + SV refer to composite models containing either two or three categories of variants. Heritability was estimated with a random effect corresponding to each category. $P$ values were calculated using two-sided Wilcoxon rank sum tests. The vertical dashed lines indicate the mean values. **b**, The proportion of heritability of traits contributed by SNPs, indels and SVs. Heritability was estimated on the basis of the SNP + indel + SV composite model (a total of 666 traits with estimated $h^2 = 0$ not shown). The numbers in parentheses represent the number of traits per group. **c**, The distribution of LD ($R^2$) between SVs and SNPs/indels within 50 kb of the SVs. For each SV, the maximum $R^2$ with adjacent SNPs/indels within 50 kb on either side is recorded. The dashed line indicates $R^2 = 0.70$. **d**, Heritability of the expression of *Solyc03G002957* contributed by *cis* and *trans* genetic variants from SL5.0-332 and TGG1.1-332. Heritability was estimated by partitioning all genetic variants into six categories (*cis*-SNPs, *cis*-indels, *cis*-SVs, *trans*-SNPs, *trans*-indels and *trans*-SVs). **e**, Manhattan plot of the expression of *Solyc03G002957* (top). The *P* value of each variant was estimated using an MLM. $n = 332$ accessions. Middle, magnification of the gene region with significant variants is shown and the dot colour represents the magnitude of LD ($R^2$) with the leading variant sv3_62128422. The circles represent SNPs and the triangles represent SVs. Genes annotated in the magnified region are shown. Bottom, LD heatmap of the magnified region. The horizontal dashed lines represent the Bonferroni threshold ($-\log_{10}[0.05/6{,}423{,}741] = 8.11$).

heritability increases to 0.37, therefore capturing an additional 0.10 of heritability (Fig. 3a). Moreover, there is still the expression of 18,174 (93.9%) genes, some with large *cis*-heritability, without any significant *cis*-eQTLs (Extended Data Fig. 7a). Our study clearly suggests that allelic heterogeneity contributes to the missing heritability of GWAS.

Multilocus models have the potential to resolve allelic heterogeneity, but only small numbers of variants can be analysed simultaneously, limiting their applications in GWAS[40]. Thus, to determine whether the graph pangenome enables capturing missing heritability by addressing allelic heterogeneity, we focused on associations between SVs within gene-proximal regions (50 kb upstream and downstream) and gene expression, motivated by the assumption that SVs are likely to be

causative. Using the least absolute shrinkage and selection operator (LASSO)[41], a multilocus regression model, we found that the expression of 1,787 out of the 19,353 genes is affected by at least two significantly associated SVs (false-discovery rate = $7.53 \times 10^{-4}$; permutation test). Compared with MLM, LASSO uniquely detected 1,249 *cis*-SV eQTLs, indicating its greater power in resolving allelic heterogeneity (Fig. 3b). The *cis*-heritability of the 1,249 eQTLs ranges from 0.00 to 0.59, with an average of 0.10. By contrast, we identified only 169 *cis*-SV QTLs with at least two significant SVs using the SL5.0-332 callset, showing the need for more thorough inclusion of genetic variants to resolve allelic heterogeneity and to capture missing heritability in GWAS. Furthermore, complex SVs such as duplications, tandem repeats and copy number

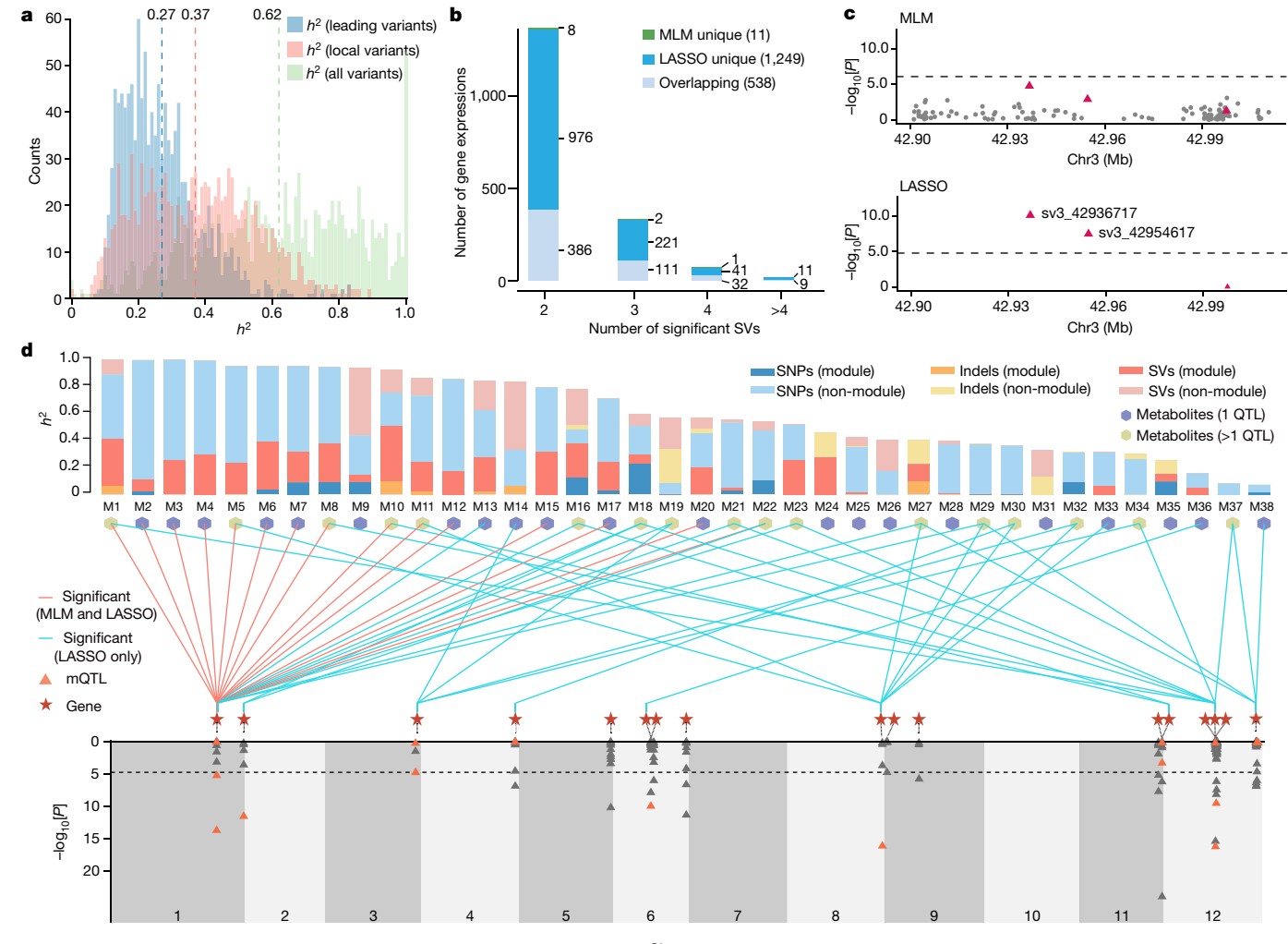

**Fig. 3 | Resolving allelic and locus heterogeneity. a**, Histogram of heritability ($h^2$) explained by leading variants within QTLs, local variants (within 50 kb on either side of the leading variants) and all genetic variants. Numbers near the vertical dashed lines represent mean $h^2$ values contributed by different variant types. Different variant types are colour coded. **b**, Allelic heterogeneity was resolved for gene expression traits. **c**, Manhattan plots for the *cis*-region (within 50 kb upstream and downstream) of the *Solyc03G001472* gene. The grey circles represent SNPs and the red triangles represent SVs. The dashed lines represent the significance thresholds. **d**, Overview of the analysis of flavonoids. Top, $h^2$ of 38 flavonoid metabolites (Supplementary Table 14), estimated with a composite model using all genetic variants from TGG1.1-332 partitioned into six different categories. 'Module' refers to variants located within 50 kb upstream or downstream of genes in the flavonoid module, and the remaining variants are 'non-module' variants. The bar plots show the contribution of each category to $h^2$, indicated by unique colours. Metabolites with more than one SV QTL are coloured in green, as indicated at the top. Statistically significant SVs for metabolites identified by both MLM and LASSO and LASSO only are coloured in red and cyan, respectively. All SVs detected by the MLM were also found by LASSO. Bottom, significant *cis*-SV eQTLs for the expression of 17 genes identified using LASSO. The 16 SVs associated with flavonoids (mQTLs) are indicated by orange triangles.

variants (CNVs), most of which are probably multiallelic SVs[36,42,43], could not be adequately addressed in this study. Thus, it is probable that allelic heterogeneity may be even more prevalent than estimated here.

By way of demonstration, we considered the gene *Solyc03G001472*, which encodes a protein of unknown function. The *cis*-heritability of this gene is 0.24 (s.d. = 0.18), contributing 52% of the total heritability. There are 646 SNPs, 46 indels and three SVs within the gene-proximal region, none of which are significantly associated with its expression when applying the MLM. Considering that the three SVs explain approximately half of the *cis*-heritability (0.12, s.d. = 0.11), we applied the LASSO model to the three SVs, and found that two of them show significant association with gene expression, one with minor allele frequency (MAF) of 0.017 (sv3_42936717) and the other with MAF of 0.032 (sv3_42954617) (Fig. 3c). The expression levels of different SV genotypes show that both SVs are associated with the expression of *Solyc03G001472* (Extended Data Fig. 7b). Overall, we show that allelic

heterogeneity can be partially addressed by cataloguing of SVs exclusively identified by the graph pangenome.

Locus heterogeneity—the phenomenon that complex traits are controlled by allelic variants across multiple genes—may also decrease the statistical power of GWAS[44]. In theory, the LASSO model could be used to resolve locus heterogeneity (as well as allelic heterogeneity) but, in practice, this is not feasible owing to the large number of genome-wide markers. An alternative approach is to focus on a network of genes potentially involved in regulating specific traits. The 'omnigenic model' postulates that all expressed genes may be involved in the regulation of complex traits[45]; however, only genes with large effects can be detected with a limited sample size. For gene expression, we derived a co-expression network formed by 99 modules, including 17,592 genes, using weighted correlation network analysis (WGCNA)[46] (Supplementary Table 8). Each module consists of an average of 177.7 genes, accounting for only 0.92% of the 19,353 expressed

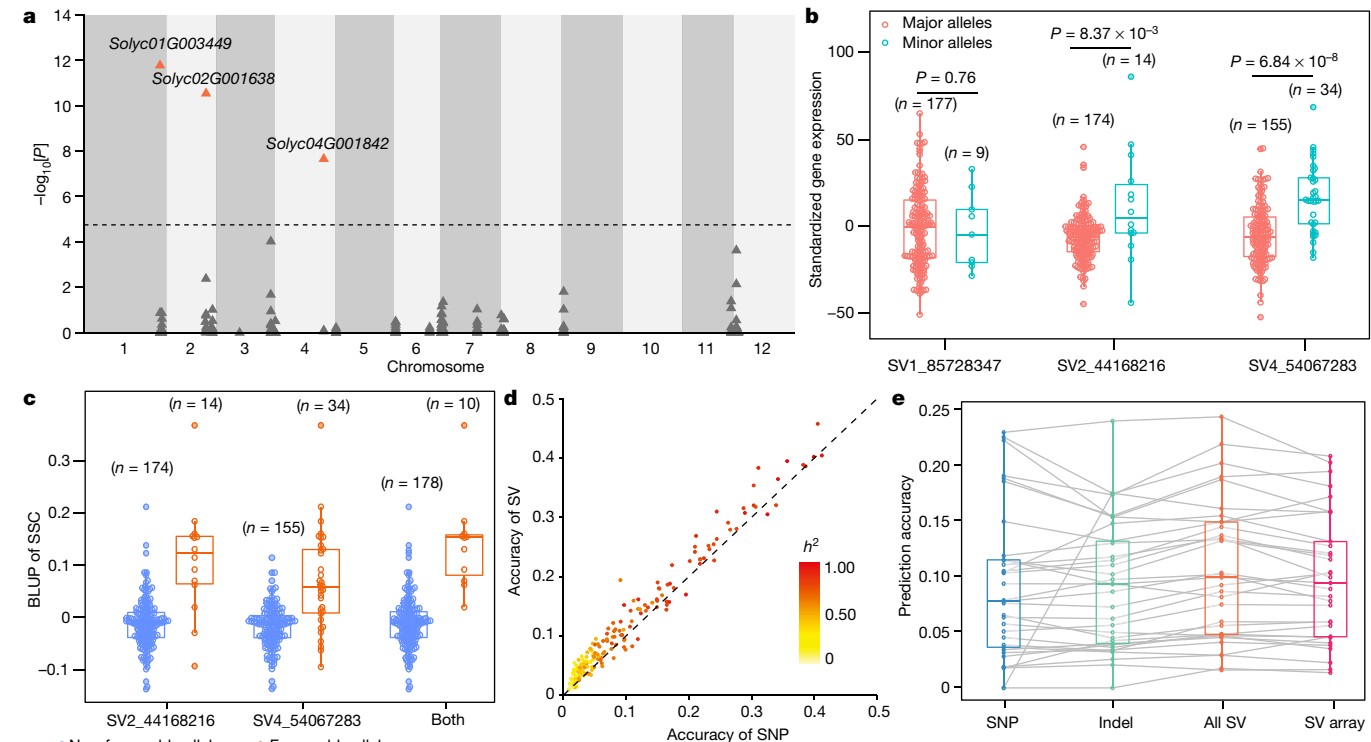

**Fig. 4 | The graph pangenome empowers MAS and genomic selection.**
**a**, Association study of SSC for the *cis*-SV set using LASSO. Genes with significant associated SVs are indicated (orange triangles). **b**, SVs affecting the expression of their nearby genes. *P* values were derived from two-sided Wilcoxon rank sum tests. **c**, Selection of accessions with high SSC using SVs. For **b** and **c**, *n* represents the sample size in each group. **d**, Comparison of genomic selection accuracies using SNPs and SVs as markers. The colour scale indicates the estimated heritability contributed by SNPs. **e**, Comparison of genomic selection accuracies using different types of genetic variants. *n* = 33 metabolic traits are plotted. 'SV array' denotes the SV panel for the DNA capture array. For **b**, **c** and **e**, the box and whisker plots show the median (centre line), mean (cross), upper and lower quartiles (box limits), 1.5× the interquartile range (whiskers) and outliers (solid points). Individual data points are plotted in circles.

genes. Notably, we found that variants within the proximal region of module genes on average contribute 0.22 of gene expression heritability, or 48.9% of the total estimated heritability (0.45) (Extended Data Fig. 7c). This indicates that genes in the same module, although fewer in number, may have disproportionately large effects on their corresponding module gene expression.

As a consequence, to address locus heterogeneity for complex traits, we can narrow the search space within a certain module in the co-expression network, and then focus on SVs affecting the corresponding gene expression. To assess the effectiveness of this strategy, we concentrated on flavonoid content (comprising 38 detected metabolites[35]), an important tomato fruit-quality trait, with heritability ranging from 0.07 to 1.00 (Fig. 3d and Supplementary Table 9). A co-expression network analysis shows that a module comprising 81 genes is related to the flavonoid pathway (hereafter, the flavonoid module) (Extended Data Fig. 8). Whole-genome SVs from TGG1.1-332 contribute on average 0.21 to the heritability of the 38 metabolite contents (range, 0.00–0.58). We found that SVs located in the proximal regions of flavonoid-module genes contribute 0.14 of heritability (Fig. 3d), suggesting that the 81 genes account for most of the genetic regulation of flavonoid content. Using LASSO, we identified 17 out of 81 genes with *cis*-SV eQTLs (Fig. 3d and Supplementary Table 10). The 171 SVs surrounding the 17 genes (*cis*-SV set) constitute the candidate dataset for evaluating the effect of locus heterogeneity on flavonoid content. We performed association analyses between the *cis*-SV set and the 38 metabolites using LASSO and identified 16 SVs surrounding nine genes associated with 31 metabolites (Supplementary Table 11). Moreover, 17 out of 31 metabolites are associated with multiple genes (Fig. 3d), suggesting that locus heterogeneity affects this complex network of flavonoids.

The nine genes affecting the 31 flavonoids consist of three genes with transcription factor activity (including the previously reported gene *SlMYB12*) and six enzyme-coding genes. In particular, Gene Ontology analysis shows that there are two transcription factors and two enzymes involved in the flavonoid biosynthetic process (Supplementary Table 12). This is one example demonstrating how the graph pangenome-based methodology sheds new light on recovering missing heritability by resolving locus heterogeneity.

## Graph pangenome empowers tomato breeding

Optimal use of the extensive genome variants is expected to facilitate a paradigm shift in crop improvement[47]. Significant genetic variants identified in GWAS are promising candidate markers for marker-assisted selection (MAS) in breeding. As a proof-of-concept study taking advantage of the added value of the graph pangenome to tomato breeding, we took fruit soluble solids content (SSC), an important yield and flavour trait, as a breeding target.

A previous study reported two QTLs underlying SSC[30], *Lin5* on chromosome 9 and *SSC11.1* on chromosome 11. To detect variants that potentially cause locus heterogeneity, we developed a universal pipeline by analysing SSC and gene expression simultaneously using WGCNA and identified a module containing 103 genes that are probably related to SSC. SVs in the proximal regions of these genes contribute 0.33 (s.d. = 0.21) to SSC heritability, comprising 52.9% of total heritability (0.62, s.d. = 0.68). Using LASSO, we identified *cis*-SV eQTLs in 25 genes among these module genes. Three SVs (SV1_85728347, SV2_44168216 and SV4_54067283) in physical proximity to the corresponding genes (*Solyc01G003449*, *Solyc02G001638* and *Solyc04G001842*) are

significantly associated with SSC (Fig. 4a). These genes are promising candidates for dissecting the genetic architecture of SSC.

Moreover, the significant genetic variants identified in this study can be valuable candidates for developing new markers to identify accessions with high SSC. We found that two of the three SVs (SV2_44168216 and SV4_54067283) significantly affect the expression of their nearby genes (*Solyc02G001638* and *Solyc04G001842*) (Fig. 4b). *Solyc02G001638* encodes a PapD-like superfamily protein, and a previous study revealed that the expression of *Solyc04G001842* encoding a trehalose-phosphate phosphatase is negatively correlated with the contents of D-fructose and D-glucose[48]. Given that SV1_85728347 is not significantly associated with the expression of *Solyc01G003449* (Fig. 4b), we did not consider this variant for MAS. We found that selecting accessions with high SSC on the basis of favourable alleles of both SV2_44168216 and SV4_54067283 is more efficient than selecting on the basis of only one SV (Fig. 4c). These results indicate that it is valuable to design marker assays with SVs, highlighting the superiority of the graph pangenome for future plant breeding.

Complex traits controlled by multiple small-effect loci limit the application of MAS in crop improvement. Genomic selection provides an alternative approach that takes advantage of small-effect QTLs. Genomic selection involves the selection of elite lines on the basis of genome-estimated breeding values from all markers, regardless of the magnitude of their effects. Using 191 metabolites of which the heritability estimated from SVs is larger than that from SNPs (0.60 versus 0.55; Wilcoxon rank sum test, $P = 0.032$), the accuracy ($r^2$ between the true phenotype and genome-estimated breeding values) of genomic selection using SVs is higher than that using SNPs (0.11 versus 0.10; Wilcoxon rank sum test, $P = 3.30 \times 10^{-32}$) (Fig. 4d). This demonstrates that capturing missing heritability using SVs improves the accuracy of genomic selection.

We next applied genomic selection to tomato flavour breeding. The estimated heritability of 33 flavour-related metabolites ranges from 0.21 to 1.00 (Supplementary Table 7). With the best linear unbiased prediction, the prediction accuracy ranges from 0.00 to 0.23, 0.00 to 0.24 and 0.02 to 0.25 using SNPs, indels and SVs, respectively, and prediction accuracy using SVs is highest for 22 of the 33 metabolites (Fig. 4e). To facilitate genomic selection in tomato breeding, we selected 20,955 candidate SVs, comprising 11,488 insertions, 9,403 deletions and 64 inversions for the design of a DNA capture array. When applied to the genomic selection of the 33 flavour-related metabolites, the SV set exhibits only limited reduction of prediction accuracy compared with the entire SV set (0.10 versus 0.11, Wilcoxon rank sum test, $P = 0.693$) (Fig. 4e). As SVs can be captured by a limited number of probes (Supplementary Note), this panel potentially provides an accurate and cost-effective platform for tomato improvement. We anticipate that future studies will validate the effectiveness of the SV array in tomato breeding. These results also enable the advancement of SV-based genomic selection in other species.

Genetic variants identified from the graph pangenome will facilitate transgenic and/or genome editing-based breeding. To improve primer design in genome editing, we designed sgRNA primers with the protospacer adjacent motif of *Cas9* for all predicted genes and released them in a web-based database (http://solomics.agis.org.cn/tomato). This database also provides tools to search the comprehensive catalogue of SNPs, indels and SVs, and to design kompetitive allele-specific PCR (KASP) markers, which can benefit the tomato research and breeding communities.

## Discussion

The state-of-the-art graph pangenome presented here incorporates genetic variants from a wide range of tomato germplasms. The inclusion of biodiversity from non-reference accessions will serve as an important platform for next-generation genomic studies and genome-assisted breeding. In particular, using the resources offered by the graph pangenome highlights the importance of SVs in capturing missing heritability by addressing incomplete LD, allelic heterogeneity and locus heterogeneity.

Here we used both read mapping and assembly-based methods to detect SVs and genotype SVs in a population using short reads using a graph-based method. One limitation is that complex SVs—for example, segmental duplications, tandem repeats and CNVs—are not specifically considered in our current pipeline. Another limitation is that only SVs present in the graph could be genotyped, and the accuracy of SV genotyping is still lower than that for SNPs and indels. Methods based on high-quality genome assemblies are superior for identifying highly complex SVs[4,49]. We believe that these problems will be addressed in the future through the development of tools that can generate a unified pangenome graph and annotation graph, reinforced by the greater availability of population-level reference-grade genome assemblies.

Some statistical tools exist that consider allelic heterogeneity, although these tools often fail to detect causal variants without high marginal $P$ values[43]. The power of these tools can probably be improved by incorporating SVs. Moreover, we have demonstrated the importance of locus heterogeneity. However, we recognize that our solution to use the LASSO is suboptimal, because it is not yet computationally feasible to consider all genetic markers at once. Ideally, multilocus tools will be developed that consider more markers. Furthermore, when it becomes feasible to genotype complex SVs, it will be necessary to develop tools that, for example, allow for multiallelic variants, and can use these variants to capture additional missing heritability and improve the accuracy of MAS and genomic selection.

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

## Methods

### Tomato sequencing and genome assembly

A total of 32 tomato accessions, including the reference cultivar Heinz 1706, were chosen from the BIG, CER and PIM groups. Genomic DNA was extracted from fresh leaves of each accession. SMRTbell libraries were constructed according to the standard protocol of PacBio (Pacific Biosciences) and sequenced on the PacBio Sequel II platform to generate HiFi reads. Primary assemblies were generated from three assemblers (Flye v.2.7, Hicanu v.2.0 and Hifiasm v.0.13)[50–52] and potential misassemblies were corrected using the GALA pipeline[53] (Supplementary Note). For the reference genome Heinz 1706, the Hi-C data were used to obtain a chromosome-level assembly. The remaining assemblies were anchored and oriented to chromosomes by the reference-guided software RagTag (v.1.0.1)[54] with the default parameters.

### Genome annotation

Protein-coding genes were predicted for each genome assembly using the MAKER2 (ref. [55]) and PRAM[56] pipelines. RNA evidence was collected by aligning RNA-sequencing (RNA-seq) reads to the repeat-masked assembly using HISAT2 (v.2.10.2)[57] and assembling them to transcripts with StringTie (v.1.3.0)[58]. TACO (v.0.7.3)[59] was applied to merge stringtie gtf (--filter-splice-juncs). Ab initio gene prediction was performed using SNAP (v.2006-07-28)[60] and AUGUSTUS (v.3.3.3)[61]. SNAP was trained for two rounds, and AUGUSTUS prediction was performed using the 'tomato' model. Proteins from SwissProt (Viridiplantae) (https://www.uniprot.org) and three *Solanum* species (*S. lycopersicum* cv. Heinz 1706 ITAG4.0 (ref. [23]), *Solanum pimpinellifolium* LA2093 (ref. [25]) and *Solanum tuberosum* DM (v.6.1)[62] were integrated, with redundant sequences removed using CD-HIT (v.4.6)[63] with the parameter '-c 0.99'. Non-redundant proteins were used for homology-based prediction using BRAKER (v.2.1.4)[64] and GeneMark (v.4.3.8)[65]. Only integrated gene models with AED values of <0.5 were retained. Furthermore, new gene models were predicted using PRAM.

### SNP and indel calling using HiFi reads

The HiFi reads were first mapped to SL5.0 using minimap2 (ref. [66]) with the parameters '-a -k 19 -O 5,56 -E 4,1 -B 5 -z 400,50 -r 2k --eqx --secondary=no'. DeepVariant (v.1.0.0) with the pretrained PacBio mode (--model_type PACBIO) was then used for variant calling of each accession, and all individual variants were merged using glnexus_cli from DeepVariant (v.0.9.0). Finally, variants that met all of the following criteria were retained: (1) total sequencing depth from 400 to 1,500; (2) quality score ≥ 20; (3) biallelic variants; (4) length ≤ 50 bp for indels.

### SV detection

To detect SVs using HiFi reads from the 31 accessions, we mapped HiFi reads to SL5.0 using NGLMR (v.0.2.7)[67] with the default parameters. A total of four callers: Sniffles (v.1.0.12)[67], SVIM (v.1.2.0)[68], CuteSV (v.1.0.10)[69] and PBSV (v.2.4.0) (https://github.com/PacificBiosciences/pbsv) with the default parameters were used for variant calling in each accession. We retained variants with a 'pass' flag and a read depth of at least three. Deletions ranging from 51 bp to 100 kb in length, and insertions ranging from 51 bp to 20 kb in length were retained. To identify SVs from the 45 genome assemblies, Assemblytics[70] was applied to the genome alignments generated using MUMmer (v.4.0)[71] with the default parameters. For the 31 accessions with SVs from the five callers, we merged all SVs shorter than 100 kb using SURVIVOR (v.1.0.6)[8] using a maximum allowed distance of 1 kb, reporting only calls supported by at least two callers and where the callers agreed regarding the type of variant. SVs longer than 100 kb detected by Assemblytics were retained. As the publicly available SVs from 100 tomatoes were identified using a different version of the reference genome (SL4.0), we transformed the coordinates to SL5.0 using the LiftOver software according to the instructions provided on the UCSC website (http://genomewiki.ucsc.edu/index.php/Minimal_Steps_For_LiftOver).

### Construction of the graph pangenome

SVs from the 31 accessions with HiFi reads and previously identified SVs from the 100 tomatoes were merged, and redundant SVs were removed according to instructions provided on GitHub (https://github.com/vgteam/giraffe-sv-paper/blob/master/scripts/sv). The variation graph toolkit (vg) pipeline[19] was used for the construction of TGG1.0, with SNPs and indels called from the HiFi reads. The vg pipeline was also used for variant calling with short reads. To obtain genotypes of variants in TGG1.0, the GBWT index was created using the greedy path-cover algorithm and 32 paths, and the default minimizer length of 29 was chosen in the minimizer index with a window size of 11. Short reads from 706 tomato accessions (>6×) were mapped to TGG1.0 with Giraffe[19] and SNPs and indels were called using DeepVariant with the NGS model. These SNPs and indels were filtered as recommended. Non-redundant SVs, SNPs and indels from both the 31 accessions with HiFi reads, the 100 accessions with ONT long reads and the 706 accessions with short reads were integrated into TGG1.1. Genotypes of SVs for the 706 accessions were called by Paragraph[18] using the default parameters.

### Graph annotation

To determine the coordinates of genes from non-SL5.0 assemblies, we calculated the distance of each accession from SL5.0 using Mash (v.2.2)[72]. We first generated a graph format for all assemblies by augmenting the 45 assemblies to SL5.0 using minigraph[73] in increasing Mash distance with the reference SL5.0, according to the instructions provided online (https://github.com/AnimalGenomicsETH/bovine-graphs). All of the coding sequences from the 45 accessions[24,25] and the previous pangenome[31] were next mapped to the graph using minigraph. Coding sequences with more than 90% coverage and sequence identity and overlapping with the SL5.0 gene models were discarded. For genes mapped to the backbone without any protein-coding gene annotation, we selected the longest one if annotated in more than one accession. For genes that were not mapped on the backbone of the graph, we removed redundant genes using CD-HIT with the parameter '-l 0.9' and only genes from the accession with the lowest distance from SL5.0 were retained. Finally, the gene sets mapped to the backbone and the graph were merged, and redundant genes were removed using CD-HIT with the parameter '-l 0.9'.

### Gene expression and metabolite contents

To quantify the expression of all genes, we used Kallisto (v.0.46.2)[74] for all 51,155 gene models in the graph pangenome. RNA-seq data from a total of 332 accessions (217 from BIG, 98 from CER and 17 from PIM) were quantified as transcripts per million (TPM). Genes with TPM values of >0.5 were defined as expressed. Only genes expressed in at least 100 accessions were retained for the downstream analyses. Raw expression levels were normalized with quantile–quantile normalization. To remove potential batch effects and confounding factors affecting gene expression, the probabilistic estimation of expression residuals method[75] was applied with the top four factors as covariates. For metabolites with missing values in <100 accessions, the mean value of two replicates was used. Raw metabolite values were transformed using the ternary logarithm and then normalized using quantile–quantile normalization.

### Heritability estimation

The LDAK-thin model[76] was used to estimate the proportion of phenotypic variance explained by genetic variants. The genetic variants were first pruned to exclude nearby SNPs in perfect LD using LDAK-thin with parameters '--window-prune 0.98 and --window-kb 100'. When computing the kinship matrix, it is necessary to specify the power parameter alpha, which determines the expected relationship between

per-variant heritability ($h_j^2$) and MAF ($f_j$). Specifically, it is assumed that $E[h_j^2]$ is proportional to $[f_j(1-f_j)]^{(1+\text{alpha})}$. By trying multiple values between −1 and 0, we found that alpha = −0.5 fits best under most scenarios, indicating a tendency for per-variant heritability to decrease with lower MAF. Principal component analysis was performed using PLINK (v.2.0)[77] using SNPs and indels from TGG1.1-332, and the first four principal components were used as covariates when estimating heritability. For partitioning contributions to heritability by different types of genetic variants, we derived the kinship for each variant category and estimated the heritability using a composite model with multiple kinship matrices. For all estimations with LDAK-thin, we added the parameter '--constrain YES' to ensure no negative estimates of heritability (if there was insufficient evidence to support the inclusion of a category, the estimated heritability was set to zero).

### Definition of heritability category

We identified the coordinates of seven anchor dots that represent the seven categories as described in Supplementary Table 13. The proportions of heritability contributed by each type of genetic variants (SNPs, indels and SVs) were used as the coordinate of each trait. Traits with heritability of zero were excluded as we could not determine the coordinate. We next calculated the Euclidean distance between the trait and each anchor dot, and each trait was assigned to the category with the shortest distance.

### Genome-wide association study

For the MLM, we used the leave-one-chromosome-out method and the mixed model implemented in GCTA[39]. After pruning using PLINK (v.2.0) with the parameter '-indep-pairwise' set to '50 5 0.2', the pruned SNPs were used for the kinship matrix (genetic relationship matrix; GRM). For SNPs and indels, the pruned dataset (-indep-pairwise 100, 1, 0.98) was used. The first four principal components were used as covariates in the model. A Bonferroni-derived threshold (0.05/total number of markers) was used as the significance threshold. For the LASSO model, the best linear unbiased prediction (BLUP) value estimate from LDAK (obtained from the composite model) was used as the response variable (new phenotype) for each trait, and the significance of genetic variants was assessed using the lassopv package[41]. The significance threshold of LASSO was determined by 1/number of SVs and the false-discovery rate at the threshold was estimated on the basis of permutations.

### QTL definition

Significant variants were grouped into the same cluster if the correlation coefficient $R^2$ of two adjacent variants was >0.20 and the physical distance was <1 Mb. Clusters containing more than three significant variants were considered as candidate QTLs. For eQTL classification, *cis*-eQTLs were inferred if the leading significant variants were <50 kb from the transcription start sites or transcription end sites of the corresponding genes; otherwise, they were considered to be *trans*-eQTLs.

### Co-expression network

WGCNA[46] was applied to the prefiltered expression data from 332 accessions to reconstruct gene modules exhibiting different expression patterns. Based on the criterion of approximate scale-free topology, the number nine was chosen as the proper soft-thresholding power for a signed network. Similar expression profiles were merged to the same module with a minimum module size set to 10 and the dissimilarity set to 0.15.

### Genomic selection

The rrBLUP[78] package was used for genomic prediction of metabolites. SNPs and indels with positive weight were used to calculate the kinship matrix with the A.mat function implemented in rrBLUP. The prediction accuracy was obtained by a five-fold cross-validation with five repetitions. As the kinship matrix was calculated from genomic data, the method is also called genomic best linear unbiased prediction.

### Reporting summary

Further information on research design is available in the Nature Research Reporting Summary linked to this paper.

## Data availability

All sequencing data generated in this study have been deposited at the Sequence Read Archive (https://ncbi.nlm.nih.gov/sra) under BioProject PRJNA733299. Whole-genome sequencing data were downloaded from NCBI (BioProjects: PRJNA259308, PRJNA353161, PRJNA454805 and PRJEB5235) and RNA-seq data were downloaded from the NCBI (BioProject: PRJNA396272). All assemblies with annotations, variant VCF files and graph files are available at the SolOmics database (http://solomics.agis.org.cn/tomato/ftp) and Sol Genomics Network (https://solgenomics.net/ftp/genomes/TGG/). The InterPro database was downloaded from https://www.ebi.ac.uk/interpro/. The UniProtKB/SwissProt database is available online (https://www.uniprot.org). Source data are provided with this paper.

## Code availability

All code associated with this project is available at GitHub (https://github.com/YaoZhou89/TGG).

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

**Acknowledgements** We thank N. Stein, M. Mascher and J. Giovannoni for comments and advice; G. Zhu for providing metabolite data; and Q. Zhang for suggestions on the estimation of heritability. This work was supported by grants from the National Natural Science Foundation of China (no. 31991180 to S.H. and no. 31801441 to Y. Zhou), the National Key Research and Development Program of China (no. 2019YFA0906200 to S.H.), the Key Research and Development Program of Guangdong Province (no. 2021B0707010005 to J.Z.), the Shenzhen Science and Technology Program (no. KQTD2016113010482651 to S.H.), the Agricultural Science and Technology Innovation Program (CAAS-ZDRW202103 to S.H.) and the US National Science Foundation (IOS-1855585 to Z.F.).

**Author contributions** S.H. and Y. Zhou conceived and designed the research. Y. Zhou, J.Z., Y.L., H. Lyu and Y.W. participated in the material preparation. S.S. and L.M. provided the Hi-C data of Heinz 1706. Y. Zhou, Z.B. and T.L. contributed to genome assembly. Z.B. contributed to genome annotation. Y. Zhou, Zhiyang Zhang and L.C. detected genetic variants. Y. Zhou and Z.B. constructed graph pangenome and annotation. Y. Zhou and Zhiyang Zhang performed gene expression and metabolites analysis. Y. Zhou, Z.B., Zhiwu Zhang, S.X. and D.S. contributed to heritability estimation and genome-wide association study. Zhiyang Zhang contributed to co-expression network analysis. Y. Zhou and Z.B. contributed to breeding analysis. Y. Zhou and Q.G. designed the SV panel of DNA capture array. K.W. and Y.W. provided metabolites and QTL data. H. Li and Y.F. contributed to computational analysis. Y. Zan, Z.F. and T.S contributed to statistical analysis. D.S., T.S., Z.F., Zhiwu Zhang, S.X., Y. Zan, Y.F. and H. Li revised the manuscript. S.H., Y. Zhou and Zhiyang Zhang wrote the manuscript. S.H. supervised the research. All of the authors read, edited and approved the manuscript.

**Competing interests** The authors declare no competing interests.

**Additional information**
**Correspondence and requests for materials** should be addressed to Sanwen Huang.

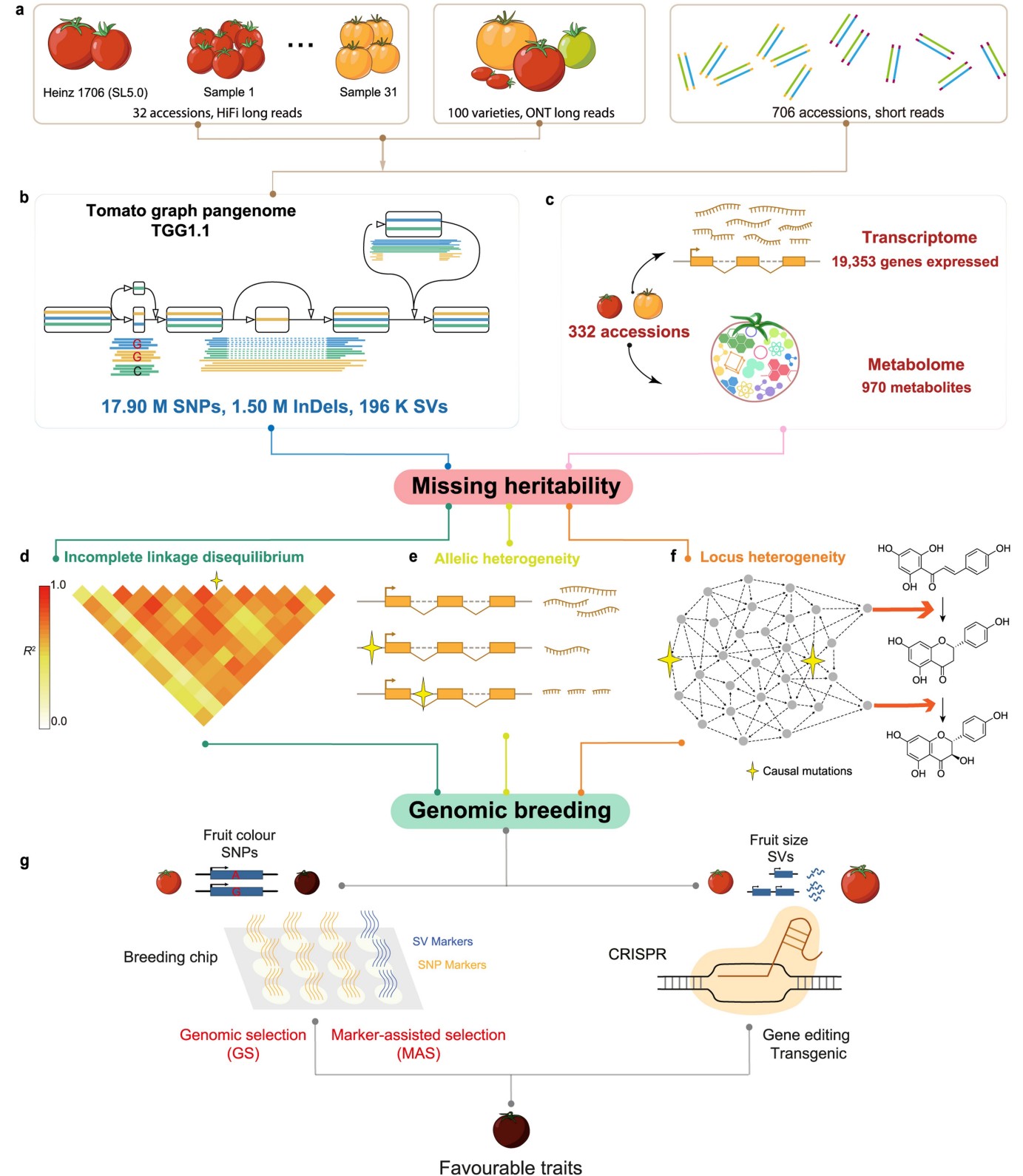

**Extended Data Fig. 1 | Layout of the tomato graph pangenome study.**
**a**) Data used for constructing the tomato graph pangenome. **b**) Sketch of the tomato graph pangenome. **c**) Profiles of metabolome and transcriptome. **d**–**f**) Potential sources of missing heritability: incomplete linkage disequilibrium (**d**), allelic heterogeneity underlying gene expression (**e**), and locus heterogeneity represented in a co-expression network (**f**). Genes affecting the different steps of the same pathway might have the same effect on the final product. Yellow stars represent causal mutations. **g**) Practical application of genomic breeding such as genomic selection (GS), marker-assisted selection (MAS) and transgenic/gene editing.

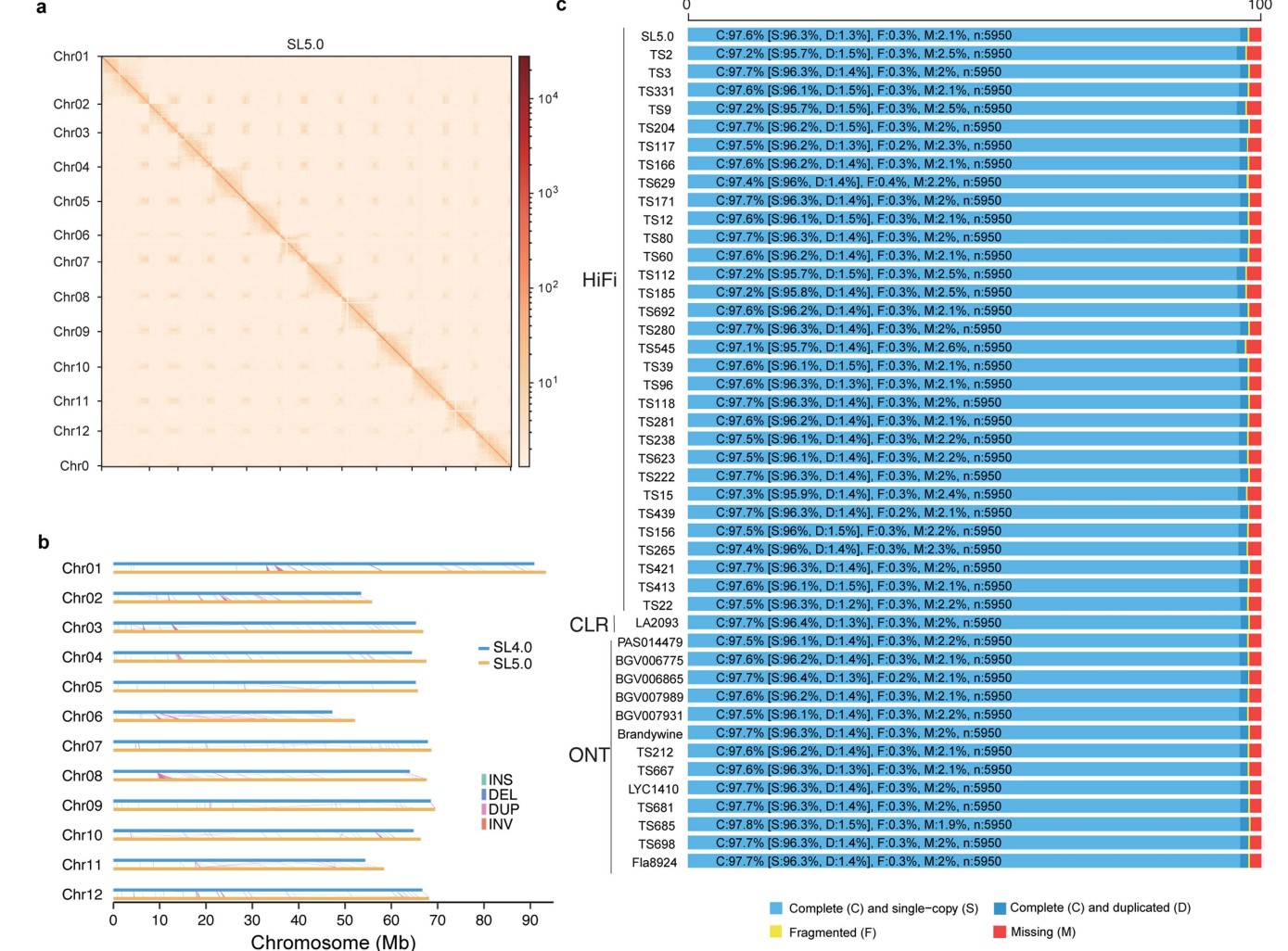

**Extended Data Fig. 2 | Characteristics of tomato genome assemblies.**
**a)** Hi-C heatmap of SL5.0. Darker red indicates higher contact probability.
**b)** Structural variants between builds SL4.0 (blue) and SL5.0 (yellow).
Insertions, deletions, duplications and inversions between SL4.0 and SL5.0

are labelled with unique colour for each type of variants. **c)** Benchmarking
Universal Single-Copy Orthologs (BUSCO) evaluation for the tomato genome
assemblies.

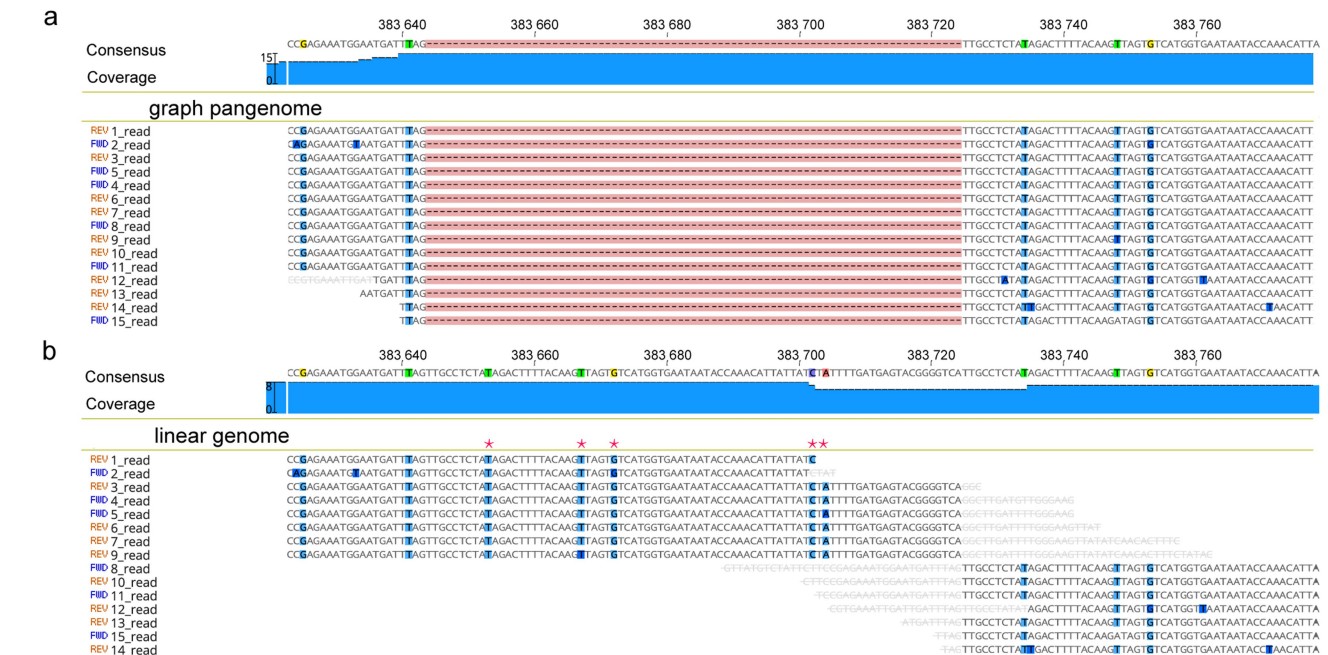

**Extended Data Fig. 3 | Read alignments to the graph pangenome and the linear genome.** Visualization of alignments of the same reads in a region by Geneious software[79] to compare differences between the graph mapper (Giraffe) (**a**) and the linear mapper (bwa) (**b**). An 81-bp deletion can be detected accurately in the graph pangenome, but soft-clipped sequences are detected in the linear genome with five false-positive SNPs (indicated by red stars).

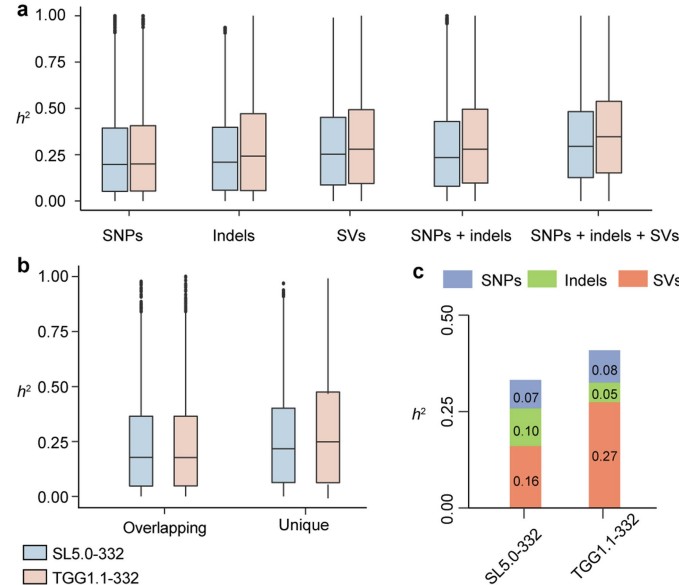

**Extended Data Fig. 4 | Evaluation of contributions to heritability by different variant types. a**) Comparison of heritability estimated from different combinations of genetic variants from SL5.0-332 and TGG1.1-332. **b**) Comparison of estimated heritability based on SNPs of different groups. n = 6,375 independent traits (**a**, **b**) were evaluated. 'Overlapping' refers to SNPs found in both TGG1.1-332 and SL5.0-332. 'Unique' refers to SNPs uniquely identified in either TGG1.1-332 or SL5.0-332. Box and whisker plots (**a**, **b**) with centre line = median, cross = mean, box limits = upper and lower quartiles, whiskers = 1.5 × interquartile range and solid points = outliers. **c**) Heritability contributed by different variant categories using a composite model.

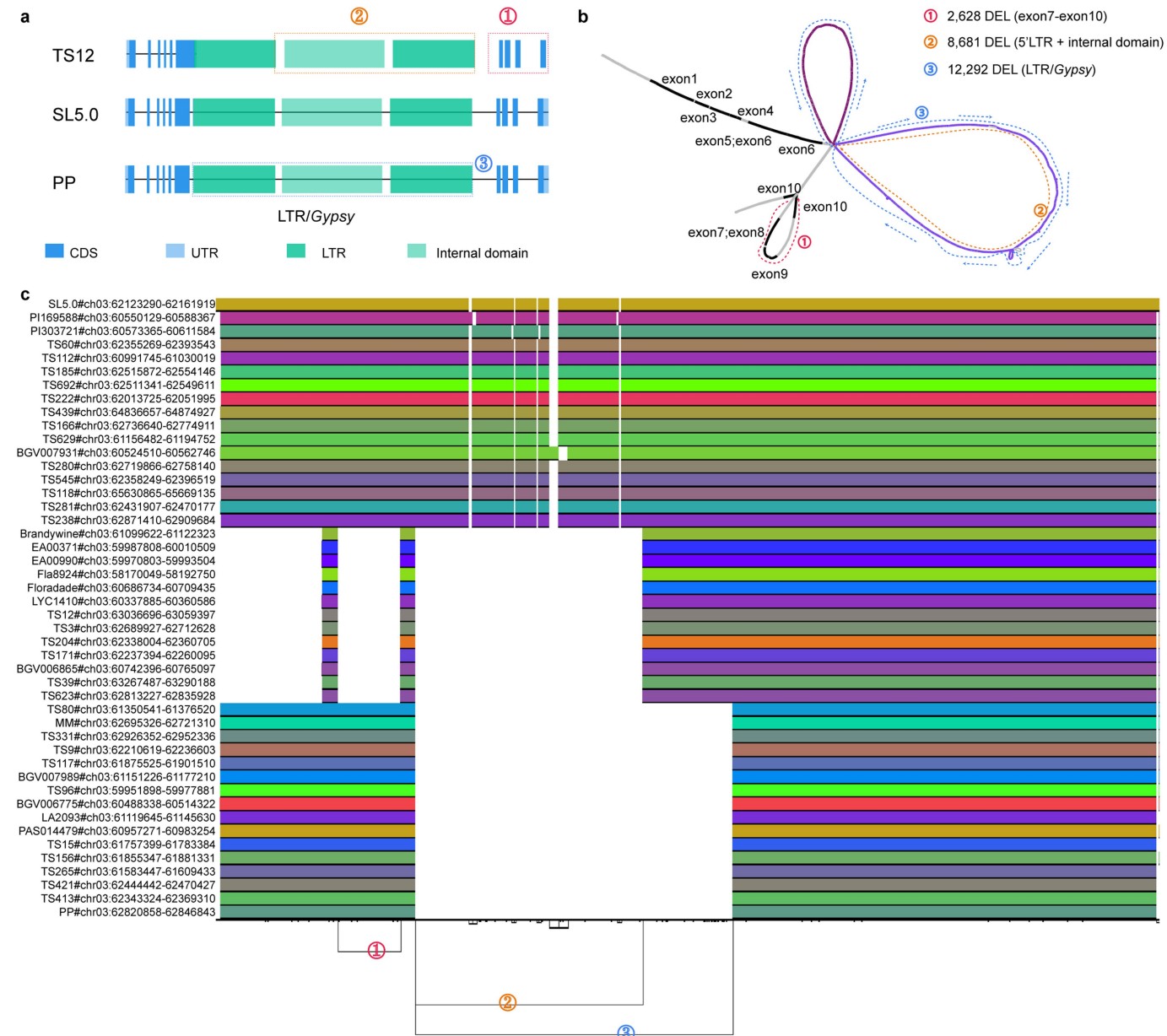

**Extended Data Fig. 5 | Gene structure of _Solyc03G002957_. a**) Different gene structures of _Solyc03G002957_; gene structures from three assemblies (TS12, SL5.0, and PP) are represented. The 2,628-bp deletion occurs at the end of the transcript. The 8,681-bp deletion in the LTR region results in a different annotation at the 3' end of the transcript. **b**) Graph representation of adjacent regions of _Solyc03G002957_. The graph was generated from the 46 assemblies shown in c). **c**) Linear representation of regions adjacent to _Solyc03G002957_.

Multiple alignment of all assemblies was performed using pggb (https://github.com/pangenome/pggb). The 8,681-bp deletion in the LTR region exists in all assemblies harbouring the haplotype with the 2,628-bp deletion. Furthermore, the multiallelic LTR deletion is represented in TGG1.1 but was filtered out in TGG1.1-332 due to low frequency, implying the potential for further improvements in genotyping multiallelic SVs using short reads.

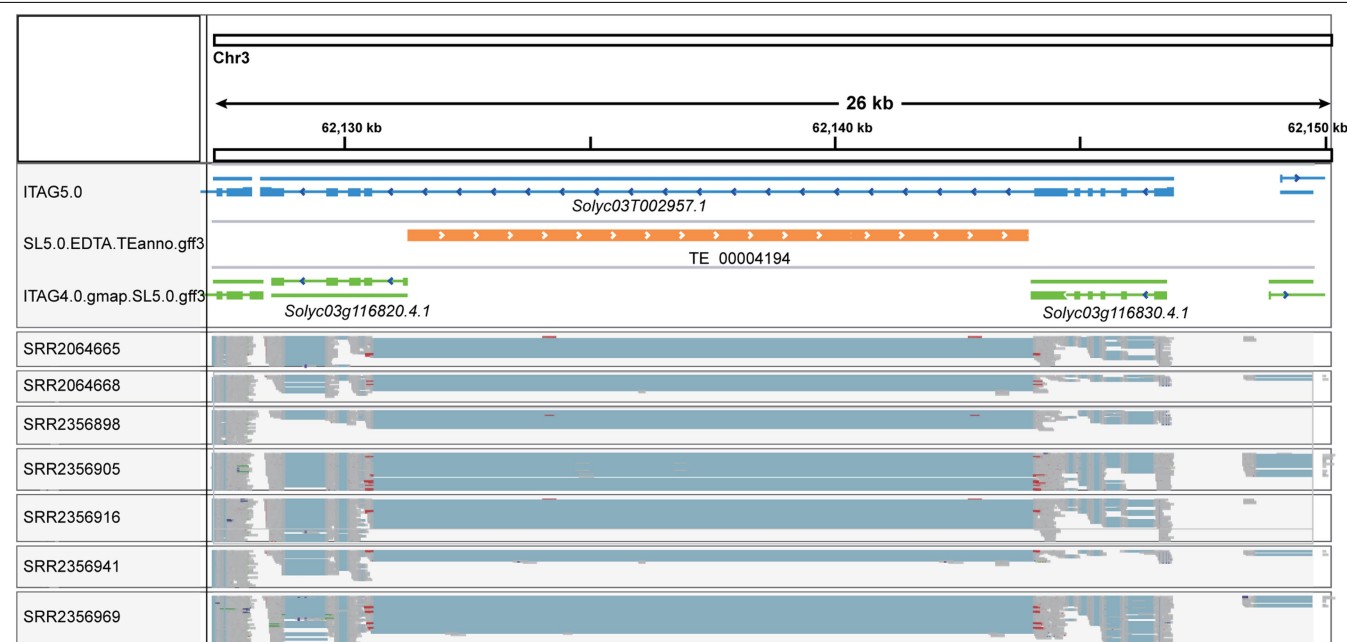

**Extended Data Fig. 6 | Integrated genome viewer of gene models according to SL5.0 and SL4.0.** This gene was misannotated as two separate genes in ITAG 4.0, possibly due to an LTR/*Gypsy* retrotransposon (12,295 bp) at the sixth intron. Blue and green lines with mRNA IDs shown represent the complete gene structure. UTRs are illustrated by thin bars, ORFs by thick bars and introns by thin lines. Arrowheads within the bars indicate transcriptional orientation. RNA-seq reads mapped to *Solyc03G002957* in SL5.0 are shown in the lower part of this figure.

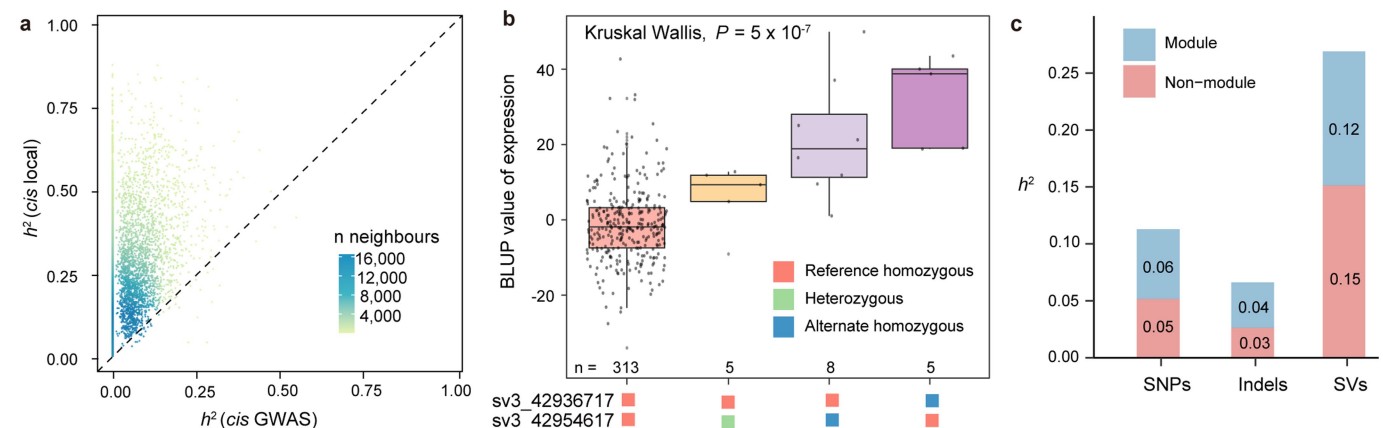

**Extended Data Fig. 7 | Allelic and locus heterogeneity in GWAS.**
**a**) Comparison of heritability estimated from leading significant variants (if present) and all genetic variants in the *cis* regions (within 50 kb upstream and downstream of a gene) for all expressed genes. If no significant variants are detected, $h^2_{cis\,gwas}$ is zero. **b**) Box plots of best linear unbiased prediction (BLUP) for the expression of *Solyc03G001472* in different genotypes of the two significant SVs. n represents number of accessions of each group. The total sample size is 331 (only groups with at least three accessions were analysed). The *P*-value was calculated from Kruskal-Wallis rank sum test. Box and whisker plots with centre line = median, cross = mean, box limits = upper and lower quartiles, whiskers = 1.5 × interquartile range and solid points = outliers. **c**) Heritability of gene expression contributed by different types of variants (SNPs, indels and SVs) within module and non-module genes in a composite model.

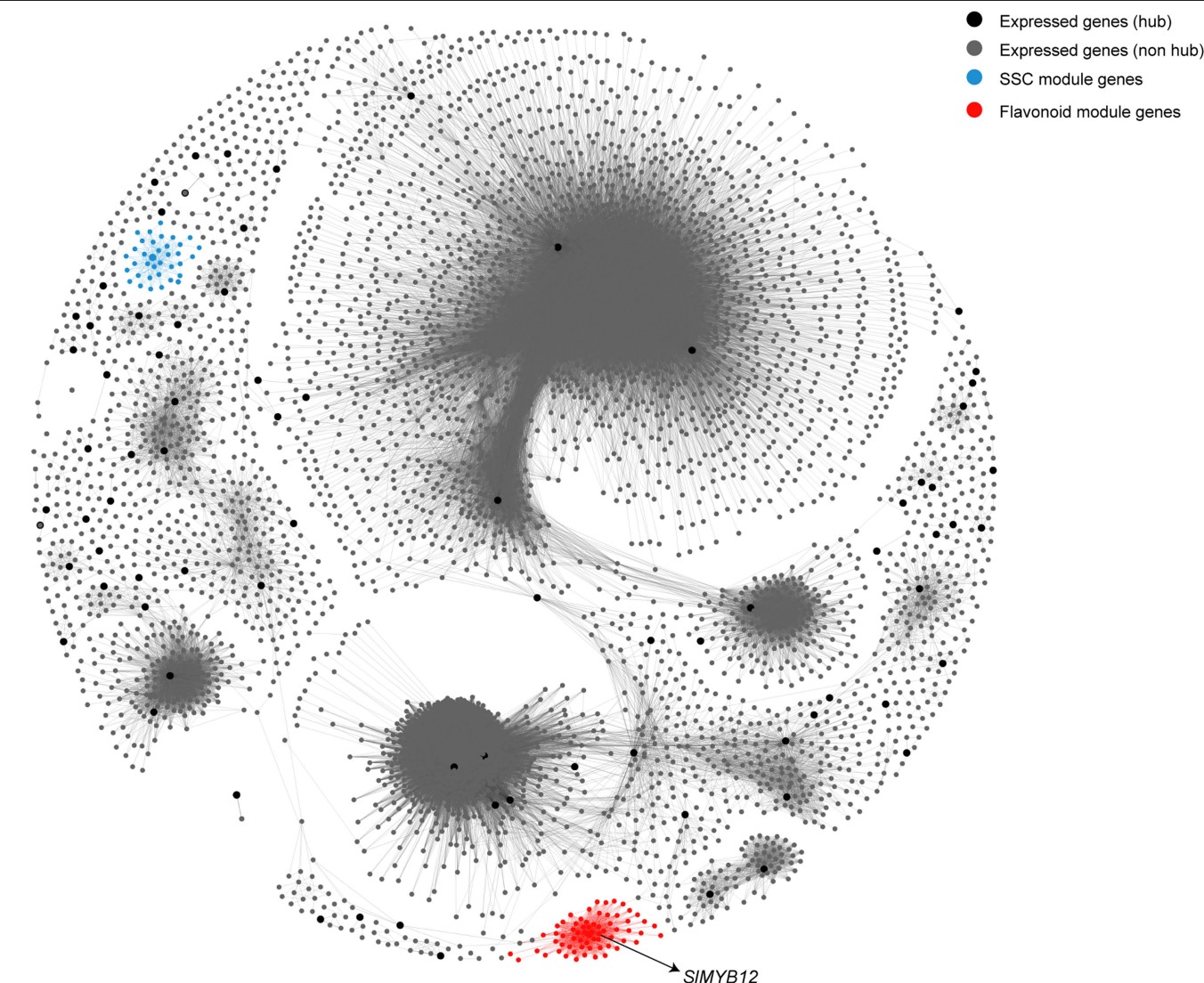

**Extended Data Fig. 8 | Co-expression network of expressed genes.** The hub genes of each module (a total of 99) are visually magnified and coloured in black and non-hub genes are coloured in grey. The soluble solids content (SSC) and flavonoid module genes are coloured in blue and red, respectively. There are 5,520 expressed genes in the network with 190,606 links (threshold > 0.05).

# Reporting Summary

## Statistics

For all statistical analyses, confirm that the following items are present in the figure legend, table legend, main text, or Methods section.

| n/a | Confirmed | |
|---|---|---|
| ☐ | ☒ | The exact sample size ($n$) for each experimental group/condition, given as a discrete number and unit of measurement |
| ☒ | ☐ | A statement on whether measurements were taken from distinct samples or whether the same sample was measured repeatedly |
| ☐ | ☒ | The statistical test(s) used AND whether they are one- or two-sided<br>*Only common tests should be described solely by name; describe more complex techniques in the Methods section.* |
| ☐ | ☒ | A description of all covariates tested |
| ☐ | ☒ | A description of any assumptions or corrections, such as tests of normality and adjustment for multiple comparisons |
| ☐ | ☒ | A full description of the statistical parameters including central tendency (e.g. means) or other basic estimates (e.g. regression coefficient) AND variation (e.g. standard deviation) or associated estimates of uncertainty (e.g. confidence intervals) |
| ☐ | ☒ | For null hypothesis testing, the test statistic (e.g. $F$, $t$, $r$) with confidence intervals, effect sizes, degrees of freedom and $P$ value noted<br>*Give P values as exact values whenever suitable.* |
| ☐ | ☒ | For Bayesian analysis, information on the choice of priors and Markov chain Monte Carlo settings |
| ☒ | ☐ | For hierarchical and complex designs, identification of the appropriate level for tests and full reporting of outcomes |
| ☒ | ☐ | Estimates of effect sizes (e.g. Cohen's $d$, Pearson's $r$), indicating how they were calculated |

*Our web collection on statistics for biologists contains articles on many of the points above.*

## Software and code

Policy information about availability of computer code

| | |
|---|---|
| Data collection | 32 tomato accessions were sequenced with High-fidelity (HiFi) long reads. |
| Data analysis | minimap2 (v2.17-r941), LDAK (v5.1), GCTA (v1.93.2), WGCNA (v1.70-3), GALA (v1.0.0), MAKER2 (v3.01.03), PRAM, LiftOver,  Paragraph(v2.2b), minigraph (v0.14-r415), rrBLUP(v4.6.1), Flye (v2.7), Hicanu (v2.0), Hifiasm (v0.13), RagTag (v1.0.1), HISAT2 (v2.10.2), StringTie (v1.3.0), TACO (v0.7.3), SNAP (v2006-07-28), Augustus (v3.3.3), CD-HIT (v4.6), BRAKER (v2.1.3), GeneMark-ET/ES (v4.3.8), DeepVariant (v0.9.0, v1.0.0), NGLMR (v0.2.7), Sniffles (v1.0.12), SVIM (v1.2.0), CuteSV (v1.0.10), PBSV(v2.4.0), SURVIVOR (v1.0.6), Mash (v2.2), Kallisto (v0.46.2), Plink (v2.0), ccs (v6.0.0), MUMmer (v4.0). The custom code is available at https://github.com/YaoZhou89/TGG. |

For manuscripts utilizing custom algorithms or software that are central to the research but not yet described in published literature, software must be made available to editors and reviewers. We strongly encourage code deposition in a community repository (e.g. GitHub). See the Nature Portfolio guidelines for submitting code & software for further information.

## Data

Policy information about availability of data

All manuscripts must include a data availability statement. This statement should provide the following information, where applicable:
- Accession codes, unique identifiers, or web links for publicly available datasets
- A description of any restrictions on data availability
- For clinical datasets or third party data, please ensure that the statement adheres to our policy

All sequence data generated in this study have been deposited at the Sequence Read Archive (https://ncbi.nlm.nih.gov/sra) under BioProject PRJNA733299. Whole-genome sequencing data was downloaded from NCBI (BioProjects PRJNA259308, PRJNA353161, PRJNA454805 and PRJEB5235) and RNA-seq data was downloaded from the NCBI (BioProject PRJNA396272). All assemblies with annotations, variants VCF files and graph files are available at SolOmics database (http://

# Field-specific reporting

Please select the one below that is the best fit for your research. If you are not sure, read the appropriate sections before making your selection.

☒ Life sciences ☐ Behavioural & social sciences ☐ Ecological, evolutionary & environmental sciences

For a reference copy of the document with all sections, see nature.com/documents/nr-reporting-summary-flat.pdf

# Life sciences study design

All studies must disclose on these points even when the disclosure is negative.

| | |
|---|---|
| Sample size | No statistical methods were used to establish sample size. The 32 accessions used for assembly were selected to represent the genetic diversity of tomato. We downloaded the public available resequencing data, RNA-seq data, and metabolomics data and accessions with all three types of data were used for association and heritability analysis. |
| Data exclusions | No data were excluded. |
| Replication | We confirmed the ability to replicate all code of this study. |
| Randomization | Randomization does not directly apply to the genome sequencing and assembly. |
| Blinding | No analyses required being blind to groups. |

# Reporting for specific materials, systems and methods

We require information from authors about some types of materials, experimental systems and methods used in many studies. Here, indicate whether each material, system or method listed is relevant to your study. If you are not sure if a list item applies to your research, read the appropriate section before selecting a response.

## Materials & experimental systems

| n/a | Involved in the study |
|---|---|
| ☒ | ☐ Antibodies |
| ☒ | ☐ Eukaryotic cell lines |
| ☒ | ☐ Palaeontology and archaeology |
| ☒ | ☐ Animals and other organisms |
| ☒ | ☐ Human research participants |
| ☒ | ☐ Clinical data |
| ☒ | ☐ Dual use research of concern |

## Methods

| n/a | Involved in the study |
|---|---|
| ☒ | ☐ ChIP-seq |
| ☒ | ☐ Flow cytometry |
| ☒ | ☐ MRI-based neuroimaging |