## [Peer Review File · Nature]

Manuscript Title: Graph pangenome captures missing heritability and empowers tomato breeding

Reviewer Comments & Author Rebuttals

Reviewer Reports on the Initial Version:

Referees' comments:

Referee #1 (Remarks to the Author):

The manuscript describes the sequencing of 32 reference level tomato genome assemblies and the construction of a graph pangenome representing structural variation across diverse individuals. The genome assemblies and pangenome graph are constructed using standard tools and current best practice. Public genome data for an additional 100 individuals was included. This represents a major resource for this species and plants in general.

The main argument of the manuscript is that the pangenome, by including genome diversity that it not represented in a single reference, captures missing heritability and provides a better reference for both association studies and genomic selection than a single reference. While this concept isn't novel and has been discussed and presented in several published pangenome papers, this work is the most comprehensive analysis of this concept, based on one of the best plant pangenomes available. Pangenome graphs are becoming increasingly available due to the increasing quality and reducing cost of long read sequence data combined with advanced bioinformatics, and these will likely become the standard for genome analysis and genomics based crop breeding, so in this context, this manuscript is foundational.

The majority of the analysis is very well done, though I do have some queries below that may aid clarity or improve the manuscript.

A major concern is that the solomics website, and hence the data, is not available and so I was unable to examine the results directly. The code linked in line 393 is also not available <https://github.com/YaoZhou89/TGG>

The authors call it a graph genome, though this may be misleading so I would recommend calling it a graph pangenome, and place this work in the context of previous plant pangenome studies.

The standard deviation for many of the statistics seem very high, given this, how confident are the authors of the numbers presented to three decimal places?

There is overuse of the word significant – there are a few sentences that are a bit hard to understand because they focus on p-values rather than on biology. For example, line 246 'Leveraging

the LASSO model, two of the three SVs, one with a minor allele frequency (MAF) of 0.017 (sv3_42936717) and the other 0.032 (sv3_42954617) were identified as significant.’ – this is hard to understand because the previous sentences talk about associations with heritability and associations with expression, it’s unclear which one is meant here.

Line 25, are the improved numbers compared to the linear genome significantly significant? Improved variant calling using a pangenome has been demonstrated previously so perhaps put this in context.

Line 115, ‘Integration of these genetic variants and the SL5.0 backbone genome resulted in the generation of an updated graph genome, which we designate as TGG1.1.’ This might be confusing to readers not used to graph genomes, who do not know that you can update the graph assembly using variants aligned with the graph. This could be clarified.

Line 189, ‘more likely’ than what?

Line 202, if the SNP is 8 genes away from the target gene, are there intervening informatics SNPs/SVs?

Line 222, is it possible to quantify the contribution of allelic heterogeneity, and is this significant?

Line 604, ‘By trying multiple values between -1 and 0, we determined alpha = -0.5 fit best, which indicates a tendency for per-variant heritability to decrease with MAF’ Details of this benchmarking should be given in supplementary

I’m pretty sure the author is Thomas Städler not Städler Thomas

One of the challenges of graph pangenomes is their visualisation and interrogation. There are some basic visualisation tools available. I don’t know if the data discussed in this manuscript could be presented using these as I cannot access the graph data.

The use of English could be improved for clarity, along with editing to correct for errors in the text.

Referee #2 (Remarks to the Author):

The manuscript entitled “Graph genome captures missing heritability and empowers tomato breeding” presents the first graph genome of tomato (TGG) and its usefulness for deciphering the genetic control of phenotypes. It shows that compared to the new reference genome they constructed which reached already a very high quality, the TGG strongly reduces the missing heritability, notably thanks to the incorporation of structural variants which provide a new type of polymorphisms. The study of allelic heterogeneity and locus heterogeneity is of particular interest using up to date methodologies.

The resource developed is of prime importance for the tomato genetic and genomic community and

should allow a new step in gene/QTL identification.

In addition to the TGG and the demonstration of its impact on traits dissection, the authors developed a chip for detecting structural variants and a catalog for gene editing. Thus this article will be a cornerstone for tomato community.

I have just a few minor remarks :

- The data are supposed to be available at the website of the authors. I could not check it as it is still private, but I think it should be available as a mirror resource on solgenomics.net which is the reference website for the Solanaceae community.
- Adding as suppl material an overview of the website and detailing ho it will be possible to add new genomes in the future would be interesting
- Most of the article is well written and very clear but the paragraph concerning SSC (L310-L365) should be corrected as many typos remain.
- Finally the SV chip information (access, details) should be increased

Referee #3 (Remarks to the Author):

The authors present a study of trait heritability in the tomato (*Solanum lycopersicum*) based on a pangenome model of the species. In advance over previous work of this type, the authors both use new sequencing data types to detect structural variants, and apply recent advances in sequence analysis to relate hundreds of other genomes to this combined model. The combination of these two technical advances yields a staggering 24% increase in the estimated heritability of 20 thousand gene expression and metabolic traits.

Specifically, the authors build a variation graph by 1) adding variants found in HiFi sequencing of 31 accessions onto a newly-assembled high-quality backbone assembly (SL5.0), to generate TGG1.0, and 2) adding variants found by mapping another 706 tomato accessions onto TGG1.0 to generate an augmented graph, TGG1.1. The authors provide ample evidence that this pangenomic reference system is sound, via validation of the assemblies and simulation experiments based on the variation graph. They then complete the alignment of 332 tomato accessions onto this graph, projecting the alignments back onto the reference genome and calling variants with DeepVariant (yielding callset TGG1.1-332). The authors then use TGG1.1-332 to study heritability versus a large panel of molecular and chemical traits. This results in the paper's key finding, that the addition of accurate structural variation calls into the set of considered genetic variants explains a large and previously mostly-unobserved portion of genetic heritability (65.9%). In conclusion, the authors meander through a number of interesting examples of where SVs contribute to traits, and include a study of the potential impact of their approach on marker-assisted and genomic selection approaches.

Overall, the work is sound and well-controlled, although I confess that I found the later sections on breeding to be thin and of less utility to the overall message. I think the authors' manuscript would be of broad interest due to its striking demonstration that SVs are such a profound contributor to heritability in this crop species. Based on this and other ongoing work, it seems likely that this result will replicate, perhaps helping to shift the focus of genomics and genome association studies away from SNPs and towards these larger and often more-impactful variants.

I have several minor comments and questions:

I found it confusing that the authors refer to the model they use as a “graph genome.” This term is ambiguous. To be more precise, it might help to specify what kind of graph model is specifically used. I would say that it’s a variation graph, or variant-based pangenome graph. Although a graph model is used, the model is equivalent to a VCF, which we clearly see in this work to be a powerful pangenome model despite its limitations. This is not clear in the main text, but it is of course clearly described in the methods text.

The authors use the graph as a coordinating system for alignment, and with the exception of a graph-based annotation pipeline, their exposition and further exploration of the data is primarily reference-based. I applaud the focus on genome graphs, but in my opinion these limiting aspects should be discussed with more care. How do the authors plan to advance the work? Could it be improved by working on complete assemblies?

In their methods the authors indicate that they keep only biallelic variants. Large sequence variants are often multiallelic. What fraction of SVs are removed by this criteria? And how specifically are segmental duplications / CNVs with multiple lengths handled? I assume that the current variant detection pipeline may not have any specific way to handle these. Do the authors have evidence that biallelic SVs like the retrotransposon insertion that they characterize are more important for heritability than copy number variants? Perhaps this study is not well-powered in this respect due to its reference-based variant detection and VCF-based pangenome graph. Given that approximately half of SVs (in many species that I am familiar with) occur in CNV hotspots, it would be very interesting to see how much more heritability could be described when including them. This of course will probably require assembly and pangenome analysis methods not available to this study. The authors should take more care in discussing the limitations of their work. Given the excellent results in terms of understanding heritability, these limitations suggest that more improvements are yet to come.

I find extended figure 5b to be somewhat confusing. The graph shows the LTR/Gypsy insertion, but it includes many apparent large bubbles that have very similar lengths of alternate alleles. What do these mean? Are they inversions or complex substitutions? The authors might want to better explain what’s happening in this figure, remove it, or add a representation of this locus in the graph that was used for their alignment and genotyping experiments. Some visual representation of e.g. TGG1.0 vs TGG1.1 might be very interesting to see.

Given the depth of the work, there is quite a lot to respond to, but in most cases I found the authors’ descriptions easy to follow, and I was left only with these minor concerns after review.

In sum this is a fascinating piece which should motivate further study in this space, and which will be of significant importance to crop breeders and geneticists of all types.

Erik Garrison

Author Rebuttals to Initial Comments:

We are delighted by the very constructive comments, and our point-by-point responses follow below. The reviewers' comments (C) are in black font, while our responses (AR) are in blue font.

We believe that our revised manuscript represents a markedly improved version with greater clarity, better balance, and improved English and readability – thanks in large part to the constructive and insightful comments offered by the three reviewers. We hope that our manuscript is now in a state that is acceptable for publication in *Nature*.

Yours sincerely,

Sanwen Huang, Ph.D.

Response to Reviewer #1

C1: The manuscript describes the sequencing of 32 reference level tomato genome assemblies and the construction of a graph pangenome representing structural variation across diverse individuals. The genome assemblies and pangenome graph are constructed using standard tools and current best practice. Public genome data for an additional 100 individuals was included. This represents a major resource for this species and plants in general. The main argument of the manuscript is that the pangenome, by including genome diversity that is not represented in a single reference, captures missing heritability and provides a better reference for both association studies and genomic selection than a single reference. While this concept isn't novel and has been discussed and presented in several published pangenome papers, this work is the most comprehensive analysis of this concept, based on one of the best plant pangenomes available. Pangenome graphs are becoming increasingly available due to the increasing quality and reducing cost of long read sequence data combined with advanced bioinformatics, and these will likely become the standard for genome analysis and genomics based crop breeding, so in this context, this manuscript is foundational.

AR1: We thank reviewer #1 for the concise summary and for supporting the scientific significance of our study.

C2: The majority of the analysis is very well done, though I do have some queries below that may aid clarity or improve the manuscript. A major concern is that the solomics website, and hence the data, is not available and so I was unable to examine the results directly. The code linked in line 393 is also not available <https://github.com/YaoZhou89/TGG>

AR2: We apologise that the website and code were not accessible during the first round of reviews. We have now made the solomics website (<http://solomics.agis.org.cn/tomato/>) available to the public. If the website is not available for any reason, you may try the IP address (<http://218.17.88.60/tomato/>). Also, we now provide a detailed document along with the code in GitHub (<https://github.com/YaoZhou89/TGG>). We also provide a mirror website for all data in the Cornell Solanaceae Genomics Network (solgenomics.net), it will be navigated in the homepage upon acceptance of the manuscript.

C3: The authors call it a graph genome, though this may be misleading so I would recommend calling it a graph pangenome, and place this work in the context of previous plant pangenome studies.

AR3: Thank you for pointing this out. Reviewer #3 also voiced the same concern and considered the term ‘graph genome’ to be ambiguous. Reviewer #3 suggested using the term variation graph considering the graph model. Therefore, we have changed the term to ‘graph pangenome’ or ‘variation graph’ throughout the manuscript, depending on the context. Briefly, we use the term variation graph when we emphasise the graph model, but in other contexts we use the term graph pangenome.

C4: The standard deviation for many of the statistics seem very high, given this, how confident are the authors of the numbers presented to three decimal places?

AR4: We agree with the reviewer that, given the large s.d. of h^2 estimated for single traits, mainly due to small population size ($n=332$), there is no good justification to report h^2 estimates to three decimal places. In the revised manuscript, these statistics are reported using two decimals. Similarly, we also changed statements such as “...56.67% of the total...” from two to one decimal place throughout the manuscript.

C5: There is overuse of the word significant – there are a few sentences that are a bit hard to understand because they focus on p-values rather than on biology. For example, line 246 ‘Leveraging the LASSO model, two of the three SVs, one with a minor allele frequency (MAF) of 0.017 (sv3_42936717) and the other 0.032 (sv3_42954617) were identified as significant.’ – this is hard to understand because the previous sentences talk about associations with heritability and associations with expression, it’s unclear which one is meant here.

AR5: We agree and have revised the manuscript accordingly. Moreover, we have changed the term “significant” to other synonyms if it does not imply statistical significance. We have revised this particular sentence to read: “Considering that the three SVs explain approximately half of the cis heritability (0.12, SD = 0.11), we applied the LASSO model to the three SVs, and found two of them show significant association with gene expression, one with minor allele frequency (MAF) of 0.017 (sv3_42936717) and the other with MAF of 0.032 (sv3_42954617)”. (Page 8, lines 254–257)

C6: Line 25, are the improved numbers compared to the linear genome significantly significant? Improved variant calling using a pangenome has been demonstrated previously so perhaps put this in context.

AR6: We believe the reviewer was concerned about line 125 (instead of line 25), which stated that the graph pangenome is better than the linear genome according to the simulation study. We apologise for not having presented the p -values. To calculate the p -values, we re-performed the simulation study multiple times. For a scenario with $10\times$ sequencing depth, the differences are significant for all types of variants (p -values are 6.30×10^{-13} , 5.04×10^{-14} and 1.69×10^{-17} for SNPs, InDels and SVs, respectively). We have revised the manuscript accordingly (Pages 4-5, lines 136–137). The p -values for all scenarios have been added in Supplementary Table S6.

C7: Line 115, ‘Integration of these genetic variants and the SL5.0 backbone genome resulted in the generation of an updated graph genome, which we designate as TGG1.1.’ This might be confusing to readers not used to graph genomes, who do not know that you can update the graph assembly using variants aligned with the graph. This could be clarified.

AR7: Thank you for this valid point! We have revised the sentence to read: “By mapping these short reads to TGG1.0, we identified additional SNPs and InDels not present in TGG1.0. After merging these variants with those from TGG1.0, we obtained a dataset comprising 17,898,731 SNPs, 1,499,161 InDels and 195,957 SVs. Integration of this updated genetic variant dataset and the SL5.0 backbone genome resulted in the generation of a new variation graph, which we designate as TGG1.1.” (Page 4, lines 122–127)

C8: Line 189, ‘more likely’ than what?

AR8: We apologise for this unclear description. We observed that the heritability of gene expression (*Solyc03G002957*) is mainly explained by SVs, indicating that SVs are more likely to be causative than SNPs/InDels. We have revised the manuscript to make this point clearer (Page 6, line 199).

C9: Line 202, if the SNP is 8 genes away from the target gene, are there intervening informatics SNPs/SVs?

AR9: Thank you for this question, which gives us a chance to provide more details for this gene. We extended the zoom-in region to 50-kb downstream of the gene (**Response Fig. 1**). We found that the $-\log_{10}(p)$ of the closest significant SNP (snp3_62171835, 24,917-bp upstream) is 9.34, which is just above the significance threshold of 8.11. Furthermore, we found variants that are not present in TGG1.1-332 (see modified Extended Figure 5a-c). We rephrased the sentence to more objectively present these results (Page 7, lines 209–210). We have also revised **Fig. 2** and **Extended Fig. 5** accordingly.

Response Figure 1. Zoom-in of GWAS signals within the gene region (chromosome 3). The darkness of the colour of each dot is proportional to the magnitude of LD (R^2) with the leading variant sv3_62128422.

C10: Line 222, is it possible to quantify the contribution of allelic heterogeneity, and is this significant?

AR10: We note that previous studies have shown that allelic heterogeneity is widespread in complex traits¹. For example, in the human GTEx dataset, 4–23% of eGenes (genes that harbor a significant eQTL) harbor allelic heterogeneity². For the human depression disorder trait, 50% of loci have strong evidence of allelic heterogeneity. Moreover, the proportion of loci with allelic heterogeneity and sample size are highly correlated ($r^2 = 0.85$). In our study, we found that expression of 1,787 out of the 19,353 genes (9.23%) was likely affected by allelic heterogeneity of SVs (page 8, line 239). Considering that our population size is relatively small (332), the proportion of genes with allelic heterogeneity may be higher if more accessions are available. We have revised the manuscript accordingly to highlight the importance of allelic heterogeneity. In addition, we discuss potential

improvements in detecting allelic heterogeneity using different models (Page 7, lines 216–219; page 12, line 389–397)

C11: Line 604, ‘By trying multiple values between -1 and 0, we determined alpha = -0.5 fit best, which indicates a tendency for per-variant heritability to decrease with MAF’ Details of this benchmarking should be given in supplementary

AR11: Thanks for pointing this out! We have added this in Supplementary Note 6. Moreover, considering that the true scenario is different from our simulations, we tried different values of alpha (-1, -0.5, -0.25, and 0) for real traits and found that the heritability of SVs is the largest, while the heritability of SNPs is the lowest under any of these scenarios (Response Fig. 2), supporting the importance of SVs in heritability estimation.

Response Figure 2. Estimation of heritability under different powers. The heritability was estimated using different types of variants under different values of power using LDAK.

C12: I'm pretty sure the author is Thomas Städler not Städler Thomas

AR12: We are quite embarrassed by this oversight and have, of course, corrected this in the revised version.

C13: One of the challenges of graph pangenomes is their visualisation and interrogation. There are some basic visualisation tools available. I don't know if the data discussed in this manuscript could be presented using these as I cannot access the graph data.

AR13: We apologise for the inaccessibility of the data during the first round of review. For visualisation, there are some tools designed for different data structures. There are native applications, such as Bandage and GfaViz or command-line tools such as vg view and odgi that can visualise the graph pangenome. For the convenience of users to use these tools, we provide the data in both VCF and vg formats on our website (<http://solomics.agis.org.cn/tomato/ftp/> and <http://218.17.88.60/tomato/ftp>). Additionally, there are some web-based tools, like Mo-MIG or Sequence Tube Map; however, they need path information in the graph. Due to the multiple sources of genetic variants (such as different callers for SVs and integration of previously identified SVs), we do not have the haplotype information in vg format, meaning that we can only know the nodes and edges in the graph, but not paths, limiting the implementation of Mo-MIG on our website. In responding to comment C9 from reviewer #3 below, we note that the pggp software could represent the region in question with haplotype information. However, although it happens to work on the particular region we are interested in, it is still challenging to generate chromosome-level alignments using pggp. In the future, we will try to generate the graph of all assemblies with help from the author of the pggp software and intend to visualise the graph by Mo-MIG on our website, which we believe will be of interest to the tomato community.

C14: The use of English could be improved for clarity, along with editing to correct for errors in the text.

AR14: In this revised version, we have made considerable efforts to improve the English throughout and to correct typos and formal incongruencies. We trust that the overall clarity and readability is now markedly enhanced.

Response to Reviewer #2

C1: The manuscript entitled “Graph genome captures missing heritability and empowers tomato breeding” presents the first graph genome of tomato (TGG) and its usefulness for deciphering the genetic control of phenotypes. It shows that compared to the new reference genome they constructed which reached already a very high quality, the TGG strongly reduces the missing heritability, notably thanks to the incorporation of structural variants which provide a new type of polymorphisms. The study of allelic heterogeneity and locus heterogeneity is of particular interest using up to date methodologies.

AR1: We thank the reviewer for the accurate and very positive summary, especially for highlighting the significance of allelic and locus heterogeneity.

C2: The resource developed is of prime importance for the tomato genetic and genomic community and should allow a new step in gene/QTL identification. In addition to the TGG and the demonstration of its impact on traits dissection, the authors developed a chip for detecting structural variants and a catalog for gene editing. Thus this article will be a cornerstone for tomato community.

AR2: Thank you for affirming the scientific significance of our manuscript. For the convenience of the tomato community, we have released all source data and summary statistics of GWAS results on our website.

I have just a few minor remarks:

C3: The data are supposed to be available at the website of the authors. I could not check it as it is still private, but I think it should be available as a mirror resource on solgenomics.net which is the reference website for the Solanaceae community.

AR3: Thank you for this suggestion! We have made our website public now, and it is available at <http://solomics.agis.org.cn/tomato> and <http://218.17.88.60/tomato>. Following your suggestion, we have also released the data at the Solgenomics website (<https://solgenomics.net/>); it will be navigated in the homepage upon acceptance of the manuscript.

C4: Adding as supplementary material an overview of the website and detailing how it will be possible to add new genomes in the future would be interesting.

AR4: We agree and have added a description of the website and the import features of the website in Supplementary Note 8. It is very interesting to discuss the scenario of adding new genomes. We note that tools for visualising, analysing, adding, deleting, and extracting information from graph pangenome are currently in their infancy. However, we expect them to come of age, and thus enable more widespread use of graph pangenomes. Currently, it is possible to augment the existing graph pangenome with new fragments, using the augment function implemented in vg. We hope that our study will inspire the development of relevant tools.

C5: Most of the article is well written and very clear but the paragraph concerning SSC (L310-L365) should be corrected as many typos remain.

AR5: We are sorry to have submitted this section with too many typos and possibly other shortcomings. We have now revised these paragraphs to make the content easier to follow. All typos have hopefully been corrected (now on lines 305–368, pages 10–11).

C6: Finally the SV chip information (access, details) should be increased

AR6: We thank the reviewer for this suggestion. We have added a new figure (Response Fig. 3, also see Fig. S18) to illustrate the SVs distribution, and the selected SVs are available at our website (<http://solomics.agis.org.cn/tomato/ftp/>). Considering the structure of our manuscript, we have put all details related to the SV chip in Supplementary Note 9.

Response figure 3. Distribution of SVs on the DNA capture array. Colours show the density within 1-Mb windows. There are 20,955 candidate SVs comprising 11,488 insertions, 9,403 deletions, and 64 inversions.

Response to Reviewer #3

C1: The authors present a study of trait heritability in the tomato (*Solanum lycopersicum*) based on a pangenome model of the species. In advance over previous work of this type, the authors both use new sequencing data types to detect structural variants, and apply recent advances in sequence analysis to relate hundreds of other genomes to this combined model. The combination of these two technical advances yields a staggering 24% increase in the estimated heritability of 20 thousand gene expression and metabolic traits.

Specifically, the authors build a variation graph by 1) adding variants found in HiFi sequencing of 31 accessions onto a newly-assembled high-quality backbone assembly (SL5.0), to generate TGG1.0, and 2) adding variants found by mapping another 706 tomato accessions onto TGG1.0 to generate an augmented graph, TGG1.1. The authors provide ample evidence that this pangenomic reference system is sound, via validation of the assemblies and simulation experiments based on the variation graph. They then complete the alignment of 332 tomato accessions onto this graph, projecting the alignments back onto the reference genome and calling variants with DeepVariant (yielding callset TGG1.1-332).

The authors then use TGG1.1-332 to study heritability versus a large panel of molecular and chemical traits. This results in the paper's key finding, that the addition of accurate structural variation calls into the set of considered genetic variants explains a large and previously mostly-unobserved portion of genetic heritability (65.9%). In conclusion, the authors meander through a number of

interesting examples of where SVs contribute to traits, and include a study of the potential impact of their approach on marker-assisted and genomic selection approaches.

AR1: We sincerely thank the reviewer for his detailed and insightful summary statement, reflecting genuine understanding of this project.

C2: Overall, the work is sound and well-controlled, although I confess that I found the later sections on breeding to be thin and of less utility to the overall message.

AR2: We appreciate this overall support and acknowledge the paucity of deep discussion on the applications to tomato breeding. We believe, however, that any advances in plant breeding methodologies could be meaningful for enhancing global food security; therefore, we are inclined to keep the appropriately revised section as part of our manuscript.

With only limited data for breeding, we still demonstrate the effectiveness of the application of the graph pangenome in both MAS and GS. GS is a fantastic way to improve the breeding process, but despite the wide use of GS in animal breeding, it is not widely used by plant breeders due to the higher cost per sample, compared with phenotypic selection or MAS. Our study shows that a carefully selected SV-set offers high prediction accuracy in GS. These SVs could be detected by a limited number of probes, reducing the cost to the level of detecting hundreds of KASP markers. This will be of interest to plant breeders. We feel sorry that, although promising in theory, the effectiveness of the SV chip in breeding has not been systematically evaluated in this study. Via our pilot study, we want to deliver the message that it might be practical to use SV-markers in plant breeding, and we will evaluate this by generating more breeding data as the next step.

C3: I think the authors' manuscript would be of broad interest due to its striking demonstration that SVs are such a profound contributor to heritability in this crop species. Based on this and other ongoing work, it seems likely that this result will replicate, perhaps helping to shift the focus of genomics and genome association studies away from SNPs and towards these larger and often more-impactful variants.

AR3: We surely agree with this sentiment and thank you for supporting this project. Our study is likely to encourage similar studies in other systems. For the convenience of readers to replicate our pipeline in their own research, we have provided a detailed document in GitHub (<https://github.com/YaoZhou89/TGG>). Each part of our analyses should be easily replicated following these guidelines.

I have several minor comments and questions:

C4: I found it confusing that the authors refer to the model they use as a “graph genome.” This term is ambiguous. To be more precise, it might help to specify what kind of graph model is specifically used. I would say that it's a variation graph, or variant-based pangenome graph. Although a graph model is used, the model is equivalent to a VCF, which we clearly see in this work to be a powerful pangenome model despite its limitations. This is not clear in the main text, but it is of course clearly described in the methods text.

AR4: Reviewer #1 has raised the same concern and we apologise for the ambiguous nature of the term ‘graph genome’. To remedy this, we have changed it to either ‘graph pangenome’ or ‘variation graph’ in our manuscript depending on the context, as described above in our AR3 to reviewer #1.

C5: The authors use the graph as a coordinating system for alignment, and with the exception of a graph-based annotation pipeline, their exposition and further exploration of the data is primarily reference-based. I applaud the focus on genome graphs, but in my opinion these limiting aspects should be discussed with more care. How do the authors plan to advance the work? Could it be improved by working on complete assemblies?

AR5: Thank you for raising these questions. We have now pointed to some limitations regarding the construction of the graph pangenome based on the available data and the scientific questions we aim to answer.

We realise that the pipeline for graph annotation is only a provisional solution as currently there is no established tool to do this. For example, we used the parameter of identity cutoff of 0.9, which likely implies that only one copy of duplicated genes in non-backbone assemblies is kept, likely affecting downstream quantification of transcripts. Although such specific examples do not affect the general conclusions of our manuscript, we highlight the importance of developing tools that are able to generate a unified pangenome graph and annotation graph. We now point this out on page 12, lines 384–387.

As shown in our study, instead of using a pruned genotype dataset, we need to generate a genotype dataset including all genetic variants, particularly in quantitative genetic studies. However, current analysis tools have limited power to circumscribe variants in complex regions, especially for complex SVs, such as the example shown below (AR9).

Genotyping multiallelic SVs is challenging. Even though we can identify SVs using assemblies, it is hard to genotype these variants precisely in a population using short reads. In addition, as we projected the variants to the backbone, we did not analyse the nested variants in SVs. As reviewer #3 stated below, large SVs are often multiallelic, and we could treat this as a specific kind of allelic heterogeneity -- it is worthwhile to try to assess the impact of these variants on SVs if we could genotype these variants correctly. The assembly-based method will be superior considering that the assemblies will be telomere-to-telomere and may yield the ability to identify complex SVs using new tools. Theoretically, all forms of structural variants can be identified given a fully contiguous and complete *de novo* assembly³. The tools for constructing a pangenome graph from assemblies are under rapid development, like cactus⁴ and pggp (<https://github.com/pangenome/pggp>).

[REDACTED]

C6: In their methods the authors indicate that they keep only biallelic variants. Large sequence variants are often multiallelic. What fraction of SVs are removed by this criteria? And how

specifically are segmental duplications / CNVs with multiple lengths handled? I assume that the current variant detection pipeline may not have any specific way to handle these.

AR6: For SV filtering, as shown in the Methods section, we followed a pipeline described at <https://github.com/vgteam/giraffe-sv-paper/blob/master/scripts/sv>. We obtained SVs from different callers for our newly sequenced 31 accessions and then merged them with SVs from previously collected 100 tomato genomes. To remove redundant SVs, we selected 20 diverse accessions (Table S16) with > 20x sequencing depth and kept alleles with supporting reads. Consequently, we apologise that we do not know how many SVs were filtered out due to possibly being multiallelic. If multiallelic SVs are kept in our dataset, they will be regarded as multiple independent SVs. For complex SVs such as segmental duplications/CNVs, we do not consider them specifically and they are likely missed by our analysis pipeline. We have revised our manuscript to point out these inherent shortcomings. (Page 12, lines 379–381)

C7: Do the authors have evidence that biallelic SVs like the retrotransposon insertion that they characterize are more important for heritability than copy number variants? Perhaps this study is not well-powered in this respect due to its reference-based variant detection and VCF-based pangenome graph.

AR7: We agree that it is currently difficult to genotype complex SVs (like segmental duplications/CNVs) using short-reads technology. There are some studies alluding to the importance of these variants⁵⁻⁷, and we agree that it is worthwhile thinking about the importance of these variants on missing heritability. We acknowledge the shortcomings and discuss this in the revised Discussion section. (Page 4, line 116; Page 8, lines 245–248; page 12, lines 394–397)

C8: Given that approximately half of SVs (in many species that I am familiar with) occur in CNV hotspots, it would be very interesting to see how much more heritability could be described when including them. This of course will probably require assembly and pangenome analysis methods not available to this study. The authors should take more care in discussing the limitations of their work. Given the excellent results in terms of understanding heritability, these limitations suggest that more improvements are yet to come.

AR8: Thank you so much for pointing out the likely importance of CNVs! Due to the limitations of our methods (as surmised by the reviewer), at present we cannot exactly evaluate the contributions of segmental duplications or CNVs to heritability.

We want to highlight the importance of these variants in an indirect way. We first identified duplications and CNVs using SyRI⁸ for all available assemblies. In summary, we identified 126 to 395 tandem repeats, 252 to 3,640 duplications, and 123 to 2,528 CNVs in each assembly. After merging the regions from all assemblies for each type of variant, we identified 2,299 regions of tandem repeats, 7,625 regions of duplications and 10,649 regions of CNVs, constituting 24.4 Mb, 158.4 Mb, and 50.4 Mb, respectively, of the tomato genome (see Response Fig. 4a). We found that there are 1,411 tandem repeats, 4,874 duplications and 6,574 CNVs that locate within 50-kb up-/down-stream of genes (Response Fig. 4b). We then analysed the enrichment of heritability across different genomic regions (Response Fig. 4c). The result shows that heritability is enriched in gene regions. Based on these observations, we infer that these complex SVs might have important effects on certain genes. Furthermore, we found that there are 531 tandem repeats, 1,931 duplications and 1,773 CNVs that overlap with the expressed genes (gene body and 2-kb upstream). These genes are

widely distributed in the co-expression network (Response Fig. 4d-f). In conclusion, we suggest that these complex SVs likely have important roles in the regulation of complex traits. As we do not have direct evidence to support this conclusion, we only revised the manuscript to discuss the limitations of our study in detecting complex SVs, together with the multiallelic SVs. (Page 12, lines 389–397)

Response Figure 4. Effect of complex structural variants. **a)** Distribution of tandem repeats, duplications, and CNVs across the genome. The position of each SV-type indicates that there are SVs in at least one assembly. **b)** Length distribution of each type of variant. **c)** Enrichment of heritability across different genomic regions. To calculate the enrichment score, we first calculated regional heritability of each trait, namely intron, exon, upstream (0, 2 kb], upstream (2 kb, 10 kb], upstream (10 kb, 20 kb], upstream (20 kb, 50 kb]; all other regions were considered as intergenic. The enrichment score represents the ratio between h^2 (target region)/number of SVs within the target region and h^2 (intergenic region)/number of SVs within an intergenic region. **d) – f)** Distribution of expressed genes that physically overlap with tandem repeats (**d**), duplications (**e**), and CNVs (**f**) in the co-expression network. Overlapping genes are colored in red.

C9: I find extended figure 5b to be somewhat confusing. The graph shows the LTR/Gypsy insertion, but it includes many apparent large bubbles that have very similar lengths of alternate alleles. What do these mean? Are they inversions or complex substitutions? The authors might want to better explain what's happening in this figure, remove it, or add a representation of this locus in the graph that was used for their alignment and genotyping experiments.

AR9: We apologize for the possible misleading presentation of this graph. We should have explained that this graph was generated by minigraph using the 46 assemblies. The bubbles represent the variations on the non-backbone path. We noticed that the output from minigraph does not represent the 2,628-bp SV clearly. Minigraph constructs a graph iteratively by mapping each assembly to an existing graph, while the software pgggb first performs an all-to-all alignment and then smoothes the graph. Therefore, we carefully looked at the region (gene *Solyc03G002957*) again using the software pgggb (-p 90 -s 100000). We found that in addition to the ~12-kb LTR deletion in some accessions, there is another ~8.6-kb deletion in the LTR region, which is in the same haplogroup as the 2,628 bp deletion (Extended Fig. 5a). We thus found that the original output from minigraph is misleading. Therefore, to better represent this region, we used the figure directly from pgggb (Extended Fig. 5b,c). Note that Extended Figs. 5a-c are all modified from the previous version.

C10: Some visual representation of e.g. TGG1.0 vs TGG1.1 might be very interesting to see.

AR10: Thank you for this suggestion! We agree that the visualisation of graph pangenomes is necessary. The node in vg is set to 32, the visualisation of the entire region is intuitively hard to understand; therefore, we only represent an InDel around gene *Solyc03G002957* (Response Fig. 5). We have added this in the Supplementary Note 4.

Response Figure 5. Visualisation of TGG1.0 and TGG1.1. The 12-bp deletion at Chr3:62107247 is specifically identified in TGG1.1.

C11: Given the depth of the work, there is quite a lot to respond to, but in most cases I found the authors' descriptions easy to follow, and I was left only with these minor concerns after review.

In sum this is a fascinating piece which should motivate further study in this space, and which will be of significant importance to crop breeders and geneticists of all types.

Erik Garrison

AR11: We thank Dr. Garrison for his very constructive suggestions and critical comments on our manuscript!

References

1. McClellan, J. & King, M.-C. Genetic heterogeneity in human disease. *Cell* **141**, 210–217 (2010).
2. Hormozdiari, F. *et al.* Widespread allelic heterogeneity in complex traits. *Am. J. Hum. Genet.* **100**, 789–802 (2017).
3. Mahmoud, M. *et al.* Structural variant calling: the long and the short of it. *Genome Biol.* **20**, 246 (2019).

4. Paten, B. *et al.* Cactus: Algorithms for genome multiple sequence alignment. *Genome Res.* **21**, 1512–1528 (2011).
5. Vences, M. D., Legendre, M., Caldara, M., Hagihara, M. & Verstrepen, K. J. Unstable tandem repeats in promoters confer transcriptional evolvability. *Science* **324**, 1213–1216 (2009).
6. Sudmant, P. H. *et al.* An integrated map of structural variation in 2,504 human genomes. *Nature* **526**, 75–81 (2015).
7. Collins, R. L. *et al.* A structural variation reference for medical and population genetics. *Nature* **581**, 444–451 (2020).
8. Goel, M., Sun, H., Jiao, W.-B. & Schneeberger, K. SyRI: finding genomic rearrangements and local sequence differences from whole-genome assemblies. *Genome Biol.* **20**, 277 (2019).

Reviewer Reports on the First Revision:

Referees' comments:

Referee #1 (Remarks to the Author):

Many thanks for the opportunity to read the revised version. Most of my comments have been addressed and I have no further suggestions. However, one issue remains. While the code is now accessible in github and the solomics website opens, there appears to be little if any relevant data on the solomics site. The Genomics button links to pages with minimal information on three assemblies but no data. The Variations page links to a search with no data and JBrowse buttons that just take you back to the home page. The Populations link fails to open. Other pages behave in a similar minimal way offering no real ability to download, view or otherwise access the data.

Referee #2 (Remarks to the Author):

As mentioned in the previous review, The manuscript entitled presents the first graph pangenome of tomato (TGG) and its usefulness for deciphering the genetic control of phenotypes. It shows that compared to the new reference genome they constructed which reached already a very high quality, the TGG strongly reduces the missing heritability, notably thanks to the incorporation of structural variants which provide a new type of polymorphisms. The study of allelic heterogeneity and locus heterogeneity is of particular interest using up to date methodologies.

The resource developed is of prime importance for the tomato genetic and genomic community and should allow a new step in gene/QTL identification.

In addition to the TGG and the demonstration of its impact on traits dissection, the authors developed a chip for detecting structural variants and a catalog for gene editing. Thus this article will be a cornerstone for tomato community.

The authors considerably improved the MS and answered clearly to all the reviewers remarks, notably they made available the website, linked it to Solgenomics website and gave access to the Github which permitted to check that all data were available and clearly annotated.

Referee #3 (Remarks to the Author):

The authors have provided an excellent response to my concerns. I thank them for their commitment to this impressive and inspiring work.

- Erik Garrison

Author Rebuttals to First Revision:

Response to Reviewer #1

C: Many thanks for the opportunity to read the revised version. Most of my comments have been addressed and I have no further suggestions. However, one issue remains. While the code is now accessible in github and the solomics website opens, there appears to be little if any relevant data on the solomics site. The Genomics button links to pages with minimal information on three assemblies but no data. The Variations page links to a search with no data and JBrowse buttons that just take you back to the home page. The Populations link fails to open. Other pages behave in a similar minimal way offering no real ability to download, view or otherwise access the data.

AR: We thank reviewer #1 for pointing these out. We have significantly improved the website and tested it thoroughly. The introduction page of Genomes has been revised for easy understanding and links to detailed information of Genomes have been added. The bugs related to variation and JBrowse have been fixed. We have set the links of Populations (and others) to the figures instead of text, and it now works appropriately. To avoid misunderstanding, we have removed some pages that do not have any data currently (like epigenetic). In addition, for the convenience of the users, we have released all data and added an introduction in the Sol Genomics Network (<https://solgenomics.net/projects/tgg>).

Response to Reviewer #2

C: As mentioned in the previous review, The manuscript entitled presents the first graph pangenome of tomato (TGG) and its usefulness for deciphering the genetic control of phenotypes. It shows that compared to the new reference genome they constructed which reached already a very high quality, the TGG strongly reduces the missing heritability, notably thanks to the incorporation of structural variants which provide a new type of polymorphisms. The study of allelic heterogeneity and locus heterogeneity is of particular interest using up to date methodologies.

The resource developed is of prime importance for the tomato genetic and genomic community and should allow a new step in gene/QTL identification.

In addition to the TGG and the demonstration of its impact on traits dissection, the authors developed a chip for detecting structural variants and a catalog for gene editing. Thus this article will be a cornerstone for tomato community.

The authors considerably improved the MS and answered clearly to all the reviewers remarks, notably they made available the website, linked it to Solgenomics website and gave access to the Github which permitted to check that all data were available and clearly annotated.

AR: We thank the reviewer #2 for the summary of our manuscript and appreciate all of the time and efforts provided.

Response to Reviewer #3

The authors have provided an excellent response to my concerns. I thank them for their commitment to this impressive and inspiring work.

- Erik Garrison

AR: We thank Erik Garrison for the time and efforts for reviewing our manuscript.